# OMNI-IML: TOWARDS UNIFIED INTERPRETABLE IMAGE MANIPULATION LOCALIZATION

**Chenfan Qu[1], Yiwu Zhong[2], Fengjun Guo[3,4], Lianwen Jin[1,4]***

[1]South China University of Technology, [2]Peking University,
[3]INTSIG Information Co., Ltd,
[4]INTSIG-SCUT Joint Lab on Document Analysis and Recognition
`202221012612@mail.scut.edu.cn, eelwjin@scut.edu.cn`

## ABSTRACT

Existing Image Manipulation Localization (IML) methods rely heavily on task-specific designs, making them perform well only on the target IML task, while joint training on multiple IML tasks causes significant performance degradation, hindering real applications. To this end, we propose **Omni-IML**, the first generalist model designed to unify IML across diverse tasks. Specifically, Omni-IML achieves generalization through three key components: (1) a **Modal Gate Encoder**, which adaptively selects the optimal encoding modality per sample, (2) a **Dynamic Weight Decoder**, which dynamically adjusts decoder filters to the task at hand, and (3) an **Anomaly Enhancement** module that leverages box supervision to highlight the tampered regions and facilitate the learning of task-agnostic features. Beyond localization, to support interpretation of the tampered images, we construct **Omni-273k**, a large high-quality dataset that includes natural language descriptions of tampered artifacts. It is annotated through our automatic, chain-of-thoughts annotation technique. We also design a simple-yet-effective interpretation module to better utilize these descriptive annotations. Our extensive experiments show that our single Omni-IML model achieves state-of-the-art performance across all four major IML tasks, providing a valuable solution for practical deployment and a promising direction of generalist models in image forensics. Our code and dataset are available at https://github.com/qcf-568/OmniIML.

## 1 INTRODUCTION

Manipulated images can pose serious risks to social media security. Despite the progress made in recent years, existing IML models are designed for individual IML tasks on specific image types (e.g., natural image or document) and usually fall short on other IML tasks. Consequently, the maintenance costs are high since every IML task requires an independent, task-specific IML model.

A naive solution is to jointly train an IML model using the data of all available IML tasks. However, joint training usually leads to an obvious performance degradation on all IML tasks, making the predictions unreliable. For example, HiFi-Net Guo et al. (2023) suffers from joint training and thus uses two different sets of model parameters for natural image IML and face IML, respectively. There are two main reasons why existing IML methods still cannot provide generalization across different IML tasks after joint training:

First, existing IML methods heavily rely on task-dependent architecture designs, training strategies etc. to detect tampering clues. These designs work well for the target IML task, but usually fall short on other IML tasks. For example, edge anomaly enhancement modules Dong et al. (2022); Yu et al. (2024) and object attention modules Wang et al. (2022a); Li et al. (2024b) have made significant progress in identifying forged natural objects. However, they can hardly work well on document images where edge artifacts are less obvious and object features are not distinct. Early frequency-

---

*Corresponding Author

vision Qu et al. (2023) fusion performs well on document images but has obvious performance degradation on natural images that cover much more noise and diversity.

Second, existing IML methods lack the design to distinguish diverse tampering features across different IML tasks. The IML task is challenging since the tampering methods are diverse and produce different subtle tampering cues. It is even harder to handle various IML tasks with a unified model. Models can be easily confused when learning to distinguish various tampering features.

To address the above issues, we propose Omni-IML, the first generalist model that can simultaneously perform well on multiple major IML tasks, as shown in Figure 1. Specifically, a Modal Gate Encoder is proposed to automatically select the optimal encoding modality for each input sample, based on the characteristics of the input image. And a Dynamic Weight Decoder is proposed to adaptively select the optimal decoder filters for each sample, assisting the generalist model to better cope with the highly diverse tampering features from multiple image types. These sample-adaptive designs can provide

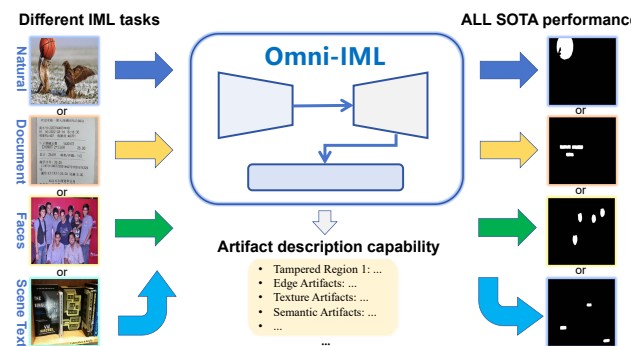

Figure 1: The proposed Omni-IML can simultaneously achieve state-of-the-art on multiple major IML tasks, without task-specific and benchmark-specific fine-tuning.

flexibility so that the model can adapt to each input sample and achieve generalization. Further, an Anomaly Enhancement is introduced to enhance the features of tampered regions with a novel box supervision design.

In addition to merely localizing tampered regions, we also seek to improve model reliability by enabling the language description of both visual and semantic artifacts of the tampered image. Despite the recent progress in interpretable IML on natural images, there still lacks a unified benchmark for interpretable IML on documents, scene text images and uncut deepfake images. The interpretable IML annotation generation pipelines with GPT-4o in previous works Xu et al. (2024); Huang et al. (2024) work well on natural images, but do not work well on documents and scene texts, where a tampered image usually contains multiple tampered regions and the image artifacts are less obvious Qu et al. (2023). There are two main reasons behind this gap: First, previous methods obtain the content, position and artifact descriptions of the tampered object with only a single query to GPT-4o. When there are multiple tampered objects in one image, GPT-4o is easily distracted and messes up the descriptions for different targets. Second, when the image artifacts are less obvious, GPT-4o's response will be less confident and even incorrect, resulting in low-quality annotations.

To this end, we propose a novel chain-of-thoughts pipeline, which solves the above issues through step-by-step focused analysis and self-examination. With the proposed method, we generate artifact descriptions for forged images from the natural, document, face, scene text IML domains, and construct a large-scale, comprehensive, high-quality dataset Omni-273k. A novel structured annotation format is adopted, which enables more reasonable and in-depth evaluation. To make better use of our Omni-273k, we further introduce a simple-yet-effective interpretation module that improves model's artifact description performance through a reference visual prompt.

We validated our methods on hundreds of representative IML tasks (e.g., IML on certificates, product photos, artwork, street photos, cards, signs, group photos, receipts, etc.), which can be categorized into four distinct **major** IML tasks, including natural image IML, document IML, face IML and scene text IML, **covering the vast majority of the recent IML research**. Extensive experiments on the four major IML tasks show that joint training the existing IML methods on all tasks leads to significant performance degradation and the trained IML models are inadequate to handle multiple IML tasks simultaneously. While our Omni-IML can minimize the performance degradation and simultaneously achieve state-of-the-art performance across all the four major IML tasks, demonstrating high scalability and effectiveness. Our main contributions are as follows:

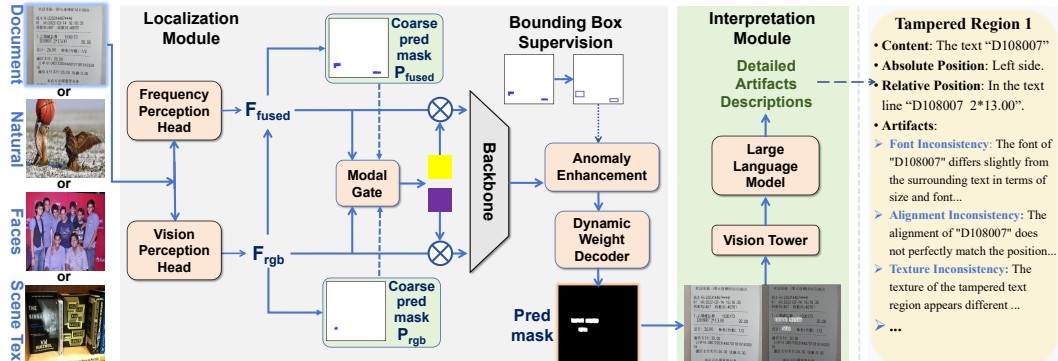

Figure 2: The overall framework of the proposed Omni-IML.

- We propose Omni-IML, **the first IML generalist model** that unifies interpretable IML across four major domains, including natural, document, face and scene text IML.
- Unified IML modeling is achieved by multiple **novel effective modules**, consisting of Modal Gate Encoder, Anomaly Enhancement, and Dynamic Weight Decoder.
- Further, to interpret image artifacts in natural language, we propose a novel chain-of-thoughts annotation technique to automatically construct a high-quality dataset Omni-273k. An effective interpretation module is also proposed to better leverage our data.
- Extensive experiments demonstrate that our **single generalist model** can simultaneously achieve **state-of-the-art** results across different major IML tasks.

## 2 RELATED WORKS

**Natural IML** aims to identify the tampered regions in daily-life style images. Existing methods leverage object-level attention Wang et al. (2022a); Li et al. (2024b), edge artifacts Dong et al. (2022); Li et al. (2023) or noise-domain filters Guillaro et al. (2023); Li et al. (2024a); Zhang et al. (2025) for better generalization. These models perform well on natural images, but mostly not well enough on other image types (e.g. documents), due to the absence of natural objects, edge and noise artifacts in these scenarios Qu et al. (2025a; 2026). Although FakeShield Xu et al. (2024) achieves image-level forensics on both natural and face images, its pixel-level forgery localization ability is still limited to natural images. Therefore, it is not the first **IML** generalist. Detailed in Appendix B.

**Document IML** aims to localize the forged regions in document images Shao et al. (2024); Dong et al. (2024), which often contain sensitive information Liu et al. (2023a); Hongliang et al. (2025). Existing methods utilize early fusion of vision and frequency features to spot unobvious artifacts Qu et al. (2023); Chen et al. (2024b). However, the model will be severely distorted in many cases of natural and face images where the frequency features are too noisy Yu et al. (2024).

**Face IML** aims to localize AI-generated fake faces. Existing methods harness metric learning Guo et al. (2023) or noise filters Liu et al. (2024) for better texture anomaly capturing. MoNFAP Miao et al. (2024). These methods show generalization on face IML but are suboptimal on natural and document images, where the tampered regions are small and the texture anomalies are unobvious.

**Scene Text IML** aims to localize tampered natural scene text in arbitrary styles and complex backgrounds. Previous works rely on specific noise patterns Wang et al. (2022b) or pre-training on authentic scene text images Qu et al. (2025b), limiting their generalization across other IML tasks.

## 3 METHODOLOGY

As shown in Figure 2, our Omni-IML has a localization module (left) to identify the tampered region, and an interpretation module (right) to describe image artifacts in natural language.

The **localization module** is roughly based on an encoder-decoder architecture. The **Modal Gate Encoder** consists of: (1) a Vision Perception Head to extract visual features from the original images;

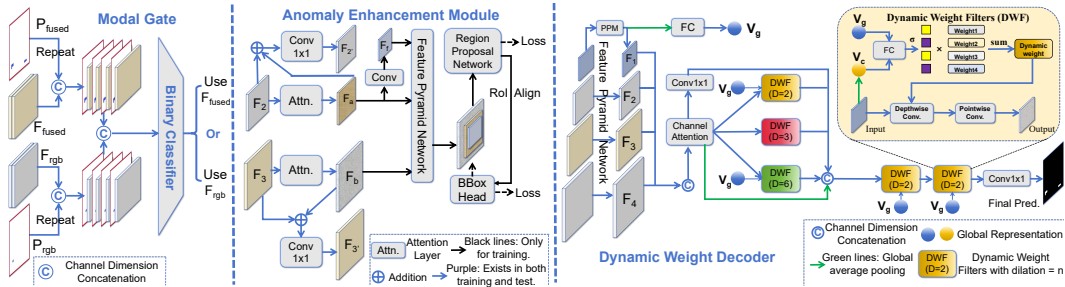

Figure 3: Modal Gate (left), Anomaly Enhancement (middle) and Dynamic Weight Decoder (right).

(2) a Frequency Perception Head to extract frequency features; (3) a Modal Gate to automatically determine the optimal modality for the following encoding process; (4) a backbone model to extract multi-scale high-level features. The **Dynamic Weight Decoder** adaptively selects the sample-wise optimal decoder filters and outputs the final mask prediction. We also design an **Anomaly Enhancement** module between the encoder and decoder, to enhance the features of tampered regions.

The **interpretation module** is an MLLM. Its input comprises both the original input image and a reference visual prompt constructed by highlighting the predicted mask on the input image.

## 3.1 MODAL GATE ENCODER

**Key Idea.** The frequency feature is a double-edged sword for the IML generalist: it can help to detect visually consistent tampering in some cases, but it may also degrade the model performance when the image is highly distorted. As a result, neither pure vision nor vision+frequency modeling can consistently provide the optimal solution. To achieve general IML through a flexible encoding modality, we propose the Modal Gate, which automatically determines the optimal encoding modality (frequency+vision or pure vision) for each input sample. The key idea of our Modal Gate Encoder is to **automatically identify the optimal modality by analyzing whether the frequency features contain too much noise, and which coarse prediction seems more confident and accurate.**

**Image Encoding**. As shown in Figure 2, we extract vision features $F_{rgb}$ using Vision Perception Head, extract frequency features using Frequency Perception Head, and obtain the fused features $F_{fused}$ by fusing $F_{rgb}$ and $F_{freq}$ with a conv-layer. The two perception heads consist of several conv-layers, with the same structure as those in previous work Qu et al. (2023). Two coarse mask predictions $P_{rgb}$ and $P_{fused}$ are obtained from $F_{rgb}$ and $F_{fused}$ with two conv-layers respectively.

**Modal Gate.** As shown in the left of Figure 3, $F_{rgb}$, $F_{fused}$, $P_{rgb}$ and $P_{fused}$ are channel-concatenated and fed into the Modal Gate for optimal modality prediction. The Modal Gate is a binary classifier consisting of several conv-layers to determine whether to use $F_{fused}$ or $F_{rgb}$ as the encoder input, by observing the noise level and the confidence of $F_{fused}$, $F_{rgb}$, and their corresponding coarse predictions $P_{rgb}$ and $P_{fused}$. We present more details in the Appendix C.1

## 3.2 ANOMALY ENHANCEMENT

Different image types from different IML tasks produce different features, joint training brings much more noise to the features and confuses the IML model. To tackle this, we propose to enhance the contrast of forged regions, and improve the feature extraction across diverse image domains through introducing a novel box supervision, as shown in the middle of Figure 3. This method harnesses task synergism to advance our IML generalist. We present more details in the Appendix C.2.

## 3.3 DYNAMIC WEIGHT DECODER

**Key Idea.** Different types of tampered images result in a wide range of manipulation clues. For example, forged objects in natural images may have abnormal contrast or edge artifacts Wang et al. (2022a), tampered text in document images might be visually consistent but has discontinuous DCT in the frequency domain Qu et al. (2023), fake faces may have unnatural texture Guo et al. (2023). These wide variations in tampering clues further cause a large variation of the encoded features of

tampered regions. Merely using a fixed set of filters for the decoder causes it to be confused by the diverse encoded features, especially in the unified training process. To this end, we propose to adaptively select the optimal decoder filters for each input image based on its characteristics. To achieve this, we propose the Dynamic Weight Decoder (DWD), as shown in the right of the Figure 3.

**Method.** In the proposed Dynamic Weight Decoder, the low-level input features are first fused with the high-level input features in a top-down manner to obtain multi-scale features $F_{1,2,3,4}$. A global feature vector $V_g$ is obtained by average pooling $F_1$. The multi-scale features are channel dimension reduced and processed by a series of Dynamic Weight Filters (DWFs) with different dilation rates. The output features are further processed by two DWFs to obtain the final prediction mask.

**Dynamic Weight Filters.** As shown in the top-right of Figure 3, to obtain the dynamic filters, we first average the input feature to obtain a current global representation $V_c$ (orange box), then interact $V_c$ with the global image vector $V_g$ (blue box) with a fully connected layer and identify the optimal dynamic filters $D_{opt}$ by weighted summation of four common convolutional filters. $A_i = \sigma(FC(V_c, V_g))$, $D_{opt} = \sum_{i=1}^{4} A_i * Wi$, where $\sigma$ is the sigmoid function, $FC$ is a linear layer, $W_i$ is the $i$th filter in the DWF. Finally, we depthwise convolve the input feature with $D_{opt}$ and then perform point-wise convolution with $1 \times 1$ conv-layer to obtain the output.

## 3.4 Interpretation Module

To provide more reliable forgery analysis by describing the image artifacts in natural language, we introduce a **simple-yet-effective** interpretation module, as illustrated on the right of Figure 2.

Existing works Xu et al. (2024) directly input the tampered image into an MLLM to describe the image artifacts. However, due to the challenging nature of image forensics, the MLLM often misidentifies the tampered region especially on multi-target and challenging scenarios (e.g. tampered documents). To address this issue, we propose to draw MLLM's attention to the suspect regions by presenting it with the forgery localization mask predicted by the mask decoder. However, directly inputting the binary mask will lead to considerable ambiguity in dense text and dense face images, since adjacent instances may have very similar positions.

To minimize the ambiguity and the difficulty in understanding the predicted tampered regions, we construct a visual reference prompt $I_{ref}$ by highlighting the mask predictions $I_{mask}$ on the tampered image $I_{input}$ by pixel-wise weighting: $I_{ref} = (I_{input} + I_{mask})/2$. We concatenate $I_{ref}$ with $I_{input}$ along the longest side, and feed it into an MLLM for text prediction. In addition to clearly indicating the suspected region, our method can also minimize overfitting and forgetting in MLLM, since it does not change the original MLLM structure.

## 4 Omni-273k Dataset

To enable the Omni-IML for high-quality description of the tampered region, we construct the Omni-273k dataset, by querying GPT-4o to generate textual artifact descriptions for the tampered regions.

### 4.1 Chain-of-Thoughts Automatic Annotation

**Motivation.** Existing works generate textual descriptions for forged objects' content, position and artifact clues in a forged image via a single prompt Xu et al. (2024); Huang et al. (2024). These methods make progress on single-target and less challenging scenarios such as natural object images, but do not work well on tampered documents, scene texts and uncut deepfakes where image artifacts are less obvious and a tampered image usually contains multiple tampered instances.

There are two main reasons for this: First, given the challenging nature of image forensics, accurately describing the content, location, and artifact of a tampered instance requires focused analysis. Simultaneous analysis of multiple targets leads to distractions, causing the model to confuse the artifacts among different objects. Second, for challenging samples such as forged documents, the response of GPT-4o is often unconfident and partially incorrect. These problems prevent previous methods from being an adequate solution for unified image forgery analysis.

**Method.** To solve the above problems and improve annotation quality. We propose a novel chain-of-thoughts pipeline, which consists of three steps, as shown in Figure 4:

**Step 1. Instance-wise Tampered Object Recognition.** In this step, we accurately recognize the content and position for each tampered region. Given the binary mask annotation $I_{mask}$ indicating the tampered

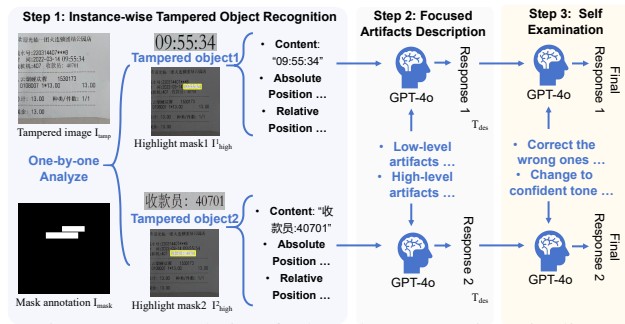

Figure 4: Our Chain-of-Thoughts annotation pipeline.

region for a tampered image $I_{tamp}$, each of the connected components in $I_{mask}$ is a tampered instance. For text images, the OCR result for each tampered instance is the content, which is obtained through an OCR engine. For other images, we highlight the $n$ tampered regions in $I_{tamp}$ respectively to get $n$ highlight masks $I_{high}^n$. We prompt GPT-4o with $I_{tamp}$ and $I_{high}^n$ sequentially for each tampered instance to get its content (e.g. "The rightmost basketball on the floor", "The face of the second young man in white shirts"). Along with recognizing the content, we also obtain the position for each tampered instance.

**Step 2. Focused Artifact Description.** In this step, we obtain detailed descriptions of image artifacts for each tampered instance. For each tampered instance, we prompt GPT-4o with $I_{tamp}, I_{high}^n$, the previously recognized content and position, and an elaborate query (lists the common artifact perspectives) to obtain the detailed descriptions $T_{des}$ for low-level visual and high-level semantic artifacts. The full query is shown in the Appendix D.1. During this artifact description process, the annotator GPT-4o is **focused** in two aspects: First, GPT-4o only focuses on one single tampered object, instead of all objects as in previous works. Second, except for the $I_{tamp}, I_{high}^n$, we also provide the previously recognized contents and positions. GPT-4o can thus focus on the task of describing image artifacts, rather than also handling multiple recognition tasks at the same time as in previous works. Consequently, the hallucination is reduced and the annotation quality is improved.

**Step 3. Self Examination.** In challenging samples where image artifacts are not obvious, the response of GPT-4o can be unconfident and incorrect, while manual filtering is costly. To this end, we propose to further improve the annotation quality by guiding GPT-4o to carefully examine and correct its previous response. For each of the tampered objects, we prompt GPT-4o with its previous response $T_{des}, I_{tamp}, I_{high}^n$, and the previously recognized contents and positions, then show GPT-4o an example containing both an unconfident, incorrect answer and the corresponding manually corrected one (shown in the Appendix D.1 in detail). With this approach, GPT-4o's final response has a considerably improved quality.

Our method differs from previous methods through **focused, step-by-step, self-examined** analysis that significantly improves annotation quality to human-level in unified IML analysis.

## 4.2 STRUCTURED ANNOTATION FORMAT

Unlike previous works that used an unstructured string as the annotation format for the artifact description, we adopt a structured JSON format, enabling much more reasonable and detailed model evaluation. Our annotation for each sample is a list of $n$ items, where $n$ is the number of tampered instances in that image. Each item is a dict with four key-value pairs:

**Tampered Region**, the value is the OCR of the tampered text or the description of the tampered object (e.g., "A sleeping orange cat").

**Absolute Position**, the value is the position of the tampered region relative to the entire image (e.g., "Top left of the image").

**Relative Position**, the value is the OCR of the text line containing the tampered text, or position of the tampered object relative to other objects (e.g., "On the leftmost green table").

**Artifacts**, the value is the artifact description of the tampered region. It is also a JSON dict, its keys are the titles for different artifacts (e.g. "Textural Artifacts", "Semantic Artifacts") and its values are the detailed descriptions under the titles.

Compared to the unstructured string format used in all previous works Xu et al. (2024); Huang et al. (2024), our **novel** structured JSON annotation offers two major advantages:

● **Reasonable Evaluation:** Different item fields require different appropriate metrics. For example, tampered text recognition and position description are close-ended tasks. These should be evaluated using exact-match metrics, such as OCR accuracy or classification accuracy. Conversely, tampered text artifact descriptions are open-ended. They necessitate evaluation with fuzzy matching metrics, like cosine similarity or ROUGE. Previous unstructured string annotations, however, only support a single metric for all item fields. This often leads to unreasonable evaluation results. For instance, entirely incorrect text recognition or position descriptions might still yield high scores under cosine similarity or ROUGE. In contrast, our structured annotation allows for the application of distinct metrics to their respective item fields, thereby producing more accurate and fair evaluation outcomes.

● **Finer-Grained Analysis:** Prior unstructured string annotations permit only very coarse model evaluation, assessing the entire output as a single unit. Our structured annotation, conversely, enables much finer-grained analysis across specific components (e.g., content, position, and artifact descriptions). Consequently, our approach can facilitate more in-depth analyses and provide significantly greater insights behind the results.

Table 1: Dataset comparison with MMTD Xu et al. (2024) and SID Huang et al. (2024). 'CM', 'SP', 'PT' are copy-move, splicing, printing. 'Manual Num.' is the number of real-world manual forgery.

| Dataset | Image Types | Tampering Methods | Target type | Forgery Num. | Manual Num. | Structured Label |
|---------|-------------|-------------------|-------------|--------------|-------------|------------------|
| MMTD | Nat.+Face | CM, SP, AIGC | Single | 21, 101 | 5, 123 | × |
| SID | Nat. | AIGC | Single | 200, 000 | 0 | × |
| **Ours** | **Nat.+Face +Doc.+S.T.** | **CM, SP PT, AIGC** | **Single and multiple** | **269, 061** | **131, 658** | ✓ |

### 4.3 DATASET CONSTRUCTION AND HIGHLIGHTS

We apply our CoT pipeline on tampered images from the commonly used datasets across the four IML domains and obtain about 273k samples. Samples and more details are presented in the Appendix E.

As shown in Table 1, our Omni-273k is **the first** one that **simultaneously covers all four major IML domains**, significantly outperforming prior works in multiple perspectives:

● **Large-Scale:** Our Omni-273k has the largest scale and $20\times$ more real-world manual forgeries than previous works, making it much more effective in evaluating real-world model performance.

● **Comprehensive:** Ours includes all the four major image domains, the most diverse tampering methods, and contains both single- and multi-target ($> 1$ tampered objects in an image) samples.

● **High-Quality:** Ours is **the first** to include structured annotation format to enable reasonable, fine-grained analysis. Ours is initially constructed and cleaned by our CoT pipeline to ensure quality.

Table 2: Comparison on natural and document IML with pixel-level IoU metric. 'DT': DocTamper.

| | Natural IML Task | | | | | | Document IML Task | | | |
|---------|--------|----------|--------|--------|-----------|------|-------------------|------|--------|--------|
| Methods | CASIA1 | Coverage | NIST16 | IMD20 | CocoGlide | Avg. | Methods | SACP | DT-Test | DT-FCD | DT-SCD |
| RRU-Net | .330 | .165 | .080 | .169 | .223 | .206 | MVSS-Net | .401 | - | - | - |
| MVSS-Net | .403 | .389 | .243 | .243 | .276 | .299 | RRU-Net | .517 | - | - | - |
| PSCC-Net | .410 | .340 | .067 | .115 | .333 | .285 | CFL-Net | .433 | - | - | - |
| CAT-Net | .684 | .238 | .238 | - | - | .406 | TIFDM | .576 | - | - | - |
| IF-OSN | .465 | .181 | .247 | .259 | .207 | .292 | MVSS-Net | - | .430 | .410 | .400 |
| EVP | .438 | .078 | .188 | .177 | .084 | .184 | PSCC-Net | - | .170 | .160 | .190 |
| TruFor | .630 | .446 | .279 | - | - | .408 | Swin-UPer | - | .700 | .410 | .510 |
| APSC-Net | **.810** | .498 | .525 | **.679** | .392 | .552 | CAT-Net | - | .710 | .600 | .540 |
| FakeShield | .540 | .357 | .320 | .500 | .523 | .448 | DTD | - | .828 | .749 | **.691** |
| SparseViT | .768 | .456 | .322 | - | .327 | .468 | SparseViT | - | .644 | .411 | .466 |
| PIM | .512 | .188 | .225 | .340 | .327 | .398 | CAFTB | - | .617 | .402 | .435 |
| Ours | .800 | **.538** | **.540** | .662 | **.538** | **.612** | Ours | **.664** | **.859** | **.863** | .676 |

Table 3: Comparison study of models trained on a specific task and all tasks. Pixel-level IoU metric.

| Method (#Model domain) | Trained on one task | | | | Trained on all tasks | | | | |
|---|---|---|---|---|---|---|---|---|---|
| | Natural | Document | SceneText | Face | Natural | Document | SceneText | Face | Average |
| CAT-Net (Natural) | .462 | .089 | .097 | .112 | .437 | .501 | .530 | .874 | .583 |
| TruFor (Natural) | .485 | .054 | .079 | .098 | .453 | .485 | .542 | .898 | .595 |
| APSC-Net (Natural) | .552 | .047 | .067 | .076 | .511 | .607 | .558 | .901 | .644 |
| SparseViT (Natural) | .483 | .042 | .081 | .080 | .450 | .499 | .529 | .886 | .591 |
| FakeShield (Natural) | .494 | .014 | .035 | .099 | .442 | .316 | .330 | .829 | .479 |
| TIFDM (Document) | .020 | .498 | .034 | .001 | .382 | .428 | .489 | .834 | .533 |
| DTD (Document) | .043 | .535 | .069 | .002 | .396 | .452 | .481 | .822 | .538 |
| Swin-UPer (SceneText) | .045 | .033 | .586 | .037 | .425 | .477 | .525 | .850 | .569 |
| UPOCR (SceneText) | .067 | .049 | .543 | .015 | .401 | .430 | .505 | .869 | .551 |
| HiFi-Net (Face) | .108 | .003 | .012 | .747 | .271 | .373 | .289 | .712 | .411 |
| MoNFAP (Face) | .090 | .022 | .004 | .902 | .398 | .467 | .499 | .845 | .552 |
| Ours (Natural) | **.628** | .063 | .092 | .104 | - | - | - | - | - |
| Ours (Document) | .061 | **.774** | .087 | .002 | - | - | - | - | - |
| Ours (SceneText) | .087 | .079 | **.623** | .030 | - | - | - | - | - |
| Ours (Face) | .058 | .006 | .008 | **.932** | - | - | - | - | - |
| Ours (All) | - | - | - | - | **.612** | **.766** | **.610** | **.923** | **.728** |

# 5 EXPERIMENTS

The localization and interpretation modules of our Omni-IML are trained totally independently. We present more implementation details in the Appendix F.

## 5.1 COMPARISON STUDY ON FORGERY LOCALIZATION

The comparison results on natural IML, document IML, face IML and scene text IML are shown in Table 2. Evidently, our generalist Omni-IML can simultaneously achieve **state-of-the-art average performance** on each individual task, demonstrating the strong generalization ability. This is because our Omni-IML can adaptively select the optimal input modality and decoder parameters for each sample, effectively producing the best features for IML on different image types. In addition, the Anomaly Enhancement module suppresses feature noise and reduces model confusion in joint training, which further improves our generalist.

To further explore the generalist capability of previous IML methods, we re-train the state-of-the-art models with their official model code, the same training data and pipeline as ours. The results are shown in the left of Table 3. In Table 3, the left part is the performance of the models trained on specific IML tasks. Evidently, all the models perform well on only one task. For example, TruFor trained on one task (Natural IML) has an IoU score of 0.485 on natural IML task but its IoU score <0.1 on other IML tasks. The right part of Table 3 is the performance of models jointly trained on all tasks. The previous IML methods suffer more performance degradation and show a much worse average performance than our Omni-IML in joint training. For example, for TIFDM, its IoU on document IML is 0.498 when it's trained on document IML task only, but its IoU on document IML is 0.428 when it's trained on all the four IML tasks, which is a degradation of 7 points. While for our Omni-IML, the IoU degradation brought by joint training is merely 0.8 points (.774 versus .766) on document IML task. This is because the previous methods rely heavily on designs and strategies targeted at one IML task, and such designs and strategies usually do not work as well on other image types (e.g. noise filters, edge enhancement and object-level attention are beneficial for natural images but not for document images). Moreover, the tampering features among diverse image types differ a lot from each other. Without a sample-adaptive and noise-suppression design, the previous methods are challenged to simultaneously learn them well. In contrast, our Omni-IML does not rely on task-dependent design and benefits from the adaptive selection of optimal encoding modality, decoder parameters, and noise-suppression. Consequently, our Omni-IML demonstrates strong generalization across different image types and has minimal performance degradation during joint training. Qualitative results for visual comparison are presented in the Appendix Figure 12.

Table 4: Comparison on Face and SceneText IML with IoU metric. 'O.F.': OpenForensics

| Face IML Task | | Scene Text IML Task | | | |
|---|---|---|---|---|---|
| Method | O.F | Methods | T-IC13 | OSTF | Avg. |
| ManTraNet | .720 | DeepLab3 | .722 | .290 | .506 |
| HPFCN | .726 | HRNetv2 | .731 | .295 | .513 |
| MVSS-Net | .701 | BEiT-UPer | .709 | .276 | .493 |
| CAT-Net | .832 | SegFormer | **.778** | .302 | .540 |
| DOAGAN | .732 | Swin-UPer | .773 | .307 | .540 |
| HiFi-Net | .749 | ConvNeXt | .776 | .310 | .543 |
| MoNFAP | .902 | UPOCR | .716 | .281 | .499 |
| Ours | **.923** | Ours | .763 | **.458** | **.610** |

Table 5: Ablation of the proposed modules on IML with IoU metric. 'w.o.': without, 'MG': Modal Gate, 'DWD': Dynamic Weight Decoder, 'AE': Anomaly Enhancement. 'Avg.': Average.

| Ablation | Natural | Document | SceneText | Face | Avg. |
|---|---|---|---|---|---|
| Baseline | .432 | .516 | .492 | .848 | .575 |
| w.o. MG | .489 | .615 | .537 | .875 | .629 |
| w.o. MG* | .538 | .634 | .561 | .904 | .659 |
| w.o. DWD | .463 | .589 | .515 | .867 | .608 |
| w.o. DW | .542 | .728 | .568 | .912 | .688 |
| w.o. AE | .548 | .707 | .570 | .916 | .682 |
| Ours | **.612** | **.766** | **.610** | **.923** | **.728** |

Table 6: Fine-grained study of image artifacts interpretation ability. 'Text Rec.': tampered text recognition, 'Abs. Pos': absolute position, 'Rel. Pos.': relative position, 'Obj. Rec.': tampered object recognition, 'Desc.': description, 'OCR': OCR accuracy. 'Acc': accuracy, 'MRB': the mean score of ROUGE-L and BLEU, 'D.S.': DeepSeek-VL, 'M.C.': MiniCPM-V-2.6, 'Int.': is InternVL3, 'Qw.': Qwen2.5-VL. 'FS.': FakeShield Xu et al. (2024), 'SI.': SIDA Huang et al. (2024).

| Method | Document | | | | | Scene Text | | | | | Face | | | | | Natural Image | | | | | Avg. |
|---|---|---|---|---|---|---|---|---|---|---|---|---|---|---|---|---|---|---|---|---|---|
| | Text. Rec. OCR | Abs. Pos. Acc | Rel. Pos. OCR | Artifacts Desc. Acc | MRB | Text. Rec. OCR | Abs. Pos. Acc | Rel. Pos. OCR | Artifacts Desc. Acc | MRB | Obj. Rec. MRB | Abs. Pos. Acc | Rel. Pos. MRB | Artifacts Desc. Acc | MRB | Obj. Rec. MRB | Abs. Pos. Acc | Rel. Pos. MRB | Artifacts Desc. Acc | MRB | MRB |
| *Supervised Fine-Tuned Models without our method* | | | | | | | | | | | | | | | | | | | | | |
| D.S.7B | .346 | .426 | .461 | .519 | .186 | .440 | .496 | .512 | .552 | .180 | .792 | .926 | .626 | .851 | .344 | .223 | .702 | .279 | .806 | .272 | .497 |
| M.C.8B | .229 | .373 | .384 | .478 | .173 | .492 | .464 | .546 | .578 | .194 | .758 | .898 | .583 | .831 | .330 | .199 | .669 | .251 | .790 | .268 | .474 |
| Int.2B | .276 | .379 | .389 | .480 | .172 | .421 | .449 | .473 | .530 | .178 | .757 | .896 | .591 | .828 | .331 | .197 | .643 | .239 | .778 | .266 | .464 |
| Qw.3B | .292 | .348 | .419 | .539 | .197 | .489 | .451 | .553 | .572 | .202 | .738 | .801 | .599 | .775 | .340 | .241 | .694 | .310 | .771 | .298 | .481 |
| Qw.7B | .312 | .381 | .429 | .521 | .202 | .580 | .536 | .640 | .614 | .217 | .768 | .879 | .630 | .794 | .342 | .270 | .752 | .342 | .749 | .301 | .512 |
| FS.13B | .153 | .325 | .275 | .407 | .160 | .382 | .347 | .429 | .405 | .153 | .706 | .765 | .556 | .730 | .303 | .181 | .635 | .226 | .732 | .253 | .406 |
| SI.13B | .154 | .328 | .275 | .411 | .160 | .383 | .349 | .430 | .405 | .153 | .706 | .765 | .557 | .730 | .303 | .182 | .635 | .226 | .732 | .253 | .406 |
| *Supervised Fine-Tuned Models with our proposed method (+Ours)* | | | | | | | | | | | | | | | | | | | | | |
| Int.2B | .610 | .595 | .654 | .648 | .233 | .615 | .586 | .639 | .693 | .226 | .770 | .938 | .587 | .847 | .339 | .241 | .746 | .292 | **.807** | .275 | .567 |
| Qw.3B | .645 | .566 | .696 | .675 | .247 | **.721** | **.619** | **.768** | .681 | .243 | .815 | .941 | .668 | .843 | .366 | **.282** | .763 | **.343** | .772 | **.310** | .598 |
| Qw.7B | **.653** | **.576** | **.698** | **.689** | **.254** | .716 | .612 | .740 | **.744** | **.252** | **.817** | **.946** | **.669** | **.862** | **.381** | .265 | .760 | .328 | .788 | .305 | .603 |

## 5.2 ABLATION STUDY ON FORGERY LOCALIZATION

The ablation results are shown in Table 5. 'w.o. MG' denotes the model without the Modal Gate and using frequency-vision fused features in encoder, it has an average IoU of .629 which is 9.9 points lower than Omni-IML of .728. This is because the frequency features in some samples are unstable, and without the Modal Gate to filter them out, these features introduce too much noise to the encoder. 'w.o. MG*' denotes the model without Modal Gate and using the pure vision modality, it has 6.9 points lower IoU than Omni-IML. This is because frequency domain modeling is also effective in some cases, especially when the tampered region is visually consistent (e.g. on document images). 'w.o. DWD' represents the model without the Dynamic Weight Decoder, it has 12.0 points lower IoU than Omni-IML. This is because the diversity of tampering features is too high for the encoder to learn them well, thus confusing the model, confirming the necessity of the proposed DWD for the generalist model. 'w.o. DW' is the model with the DWD structure but the filter weights in the decoder keep all the same for each input, it has 4.0 points lower IoU than Omni-IML, this verifies that the adaptive selection of optimal decoder weights for each sample can reduce confusion in joint training. 'w.o. AE' is the model without the proposed Anomaly Enhancement (AE) module, it has 4.6 points lower IoU than Omni-IML. This is because the proposed AE module can enhance the forged regions and suppress noise in the features. The model without any of the proposed modules serves as the 'Baseline' model, its IoU is 15.3 points lower than Omni-IML. These results have proven the effectiveness of our proposed methods.

### 5.3 EXPERIMENTS ON FORGERY INTERPRETATION

The experiments on interpretation task are conducted on our Omni-273k dataset. Unlike previous works that use one metric to evaluate the entire output string, we use our structured textual labels to perform fine-grained evaluation. As shown in Table 6, we use OCR accuracy Zhang et al. (2019) to evaluate the descriptions of tampered content and relative position for text images, and use Mean of ROUGE-L and BLEU (MRB) for non-text images. The description answers for absolute position (e.g. "Top left") and artifacts title (e.g. "Edge Artifacts") are close-ended, they can be regarded as single-choice and multi-choice tasks respectively, so we use accuracy to evaluate the ratio of matches. The detailed artifact description are open-ended, and again we use the MRB score.

By comparing the model fine-tuned with and without our method in Table 6, we can learn that our visual prompt improves model's forgery interpretation ability via reducing the misidentification of tampered regions. For example, for the Qwen2.5-VL 7B fine-tuned without our method, its description scores for tampered text, absolute and relative positions are .312, .381, .429 respectively on documents. These low scores mean that the models often incorrectly detect the tampered regions. In contrast, the same model fine-tuned with our method gets the three much higher scores of .653, .576, .698 respectively. This confirms that our visual prompt assists the model in correctly identifying the tampered region, and consequently the artifact description is much more accurate (e.g. .689 versus .521). Similar conclusion to other image types such as scene texts and faces.

FakeShield and SIDA are not highly effective. This is because their frozen SAM and LISA Lai et al. (2024) paradigms are challenged in text and multi-target scenarios Xia et al. (2024). In addition, their base MLLM LLaVA is relatively outdated with a too-small fixed input resolution of 336 and is poor at recognizing non-English text. In contrast, our method effectively improves various base MLLMs and consistently yields high performance, demonstrating high generality. Robustness evaluation and qualitative comparison are presented in the Appendix Table 13 and Figures 15, 16, 17.

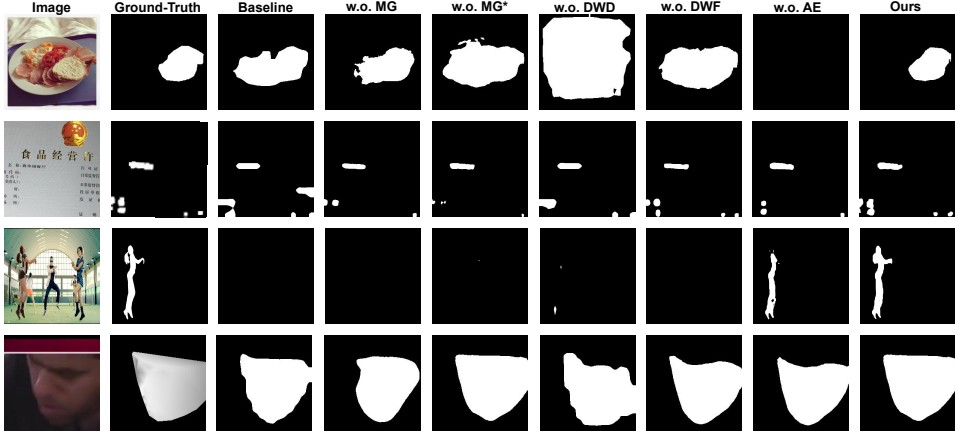

Figure 5: Qualitative ablation of the proposed modules.

## 6 CONCLUSION

We propose Omni-IML, the first generalist model for Image Manipulation Localization (IML). Our generalist model achieves generalizable localization through several novel and effective modules. These include a Modal Gate Encoder, a Dynamic Weight Decoder, and an Anomaly Enhancement module. We also introduce an effective interpretation module for a better interpretation of image artifacts in natural language. Furthermore, we constructed Omni-273k, a large-scale, comprehensive, and high-quality dataset. Its textual annotations are initially curated using a novel Chain-of-Thoughts automatic pipeline. This significantly improves annotation quality and reduces manual cleaning costs. The dataset's structured annotation also enables reasonable evaluation and in-depth analyses. Extensive experiments on four major IML tasks demonstrate that our single model achieves state-of-the-art average performance across all tasks simultaneously. We believe that this work offers valuable insights for real-world application and future research in IML.

ACKNOWLEDGMENT

This research is supported in part by the National Natural Science Foundation of China (Grant No.:62476093).

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

# A APPENDIX CONTENTS

# B DETAILED COMPARISON WITH FAKESHIELD

Although FakeShield Xu et al. (2024) also seeks to achieve explainable IML, our work differs significantly from FakeShield as following:

**1. IML Image Domains**. Our Omni-IML can effectively localize forged region in **pixel-level** across the four major IML domains, including natural IML, document IML, face IML and scene text IML. In contrast, although FakeShield is designed to identify forged natural and face images at **image-level**, its **pixel-level** forgery localization capability is still limited to a single natural image domain Xu et al. (2024). In addition, FakeShield cannot identify forged document or scene text images neither at image-level nor pixel-level. Therefore, FakeShield is not the first **pixel-level** image manipulation **localization** generalist. In contrast, ours is.

**2. Multi-target Localization Capability**. Our Omni-IML adopts a novel paradigm that prompts MLLM with the localization model's mask prediction. Therefore, ours is still effective when multiple forged regions exist in an image. In contrast, FakeShield adopts the LISA paradigm Lai et al. (2024), which is well-known to be poor in multi-target scenario Rasheed et al. (2024); Xia et al. (2024). However, multi-target forgeries scenario is common in real-world, especially for text images Luo et al. (2025); Qu et al. (2023; 2025b). This limits the application value of FakeShield.

**3. Generalist Design**. Our proposed Modal Gate, Anomaly Enhancement and Dynamic Weight Decoder are typically designed to alleviate performance degradation in joint task training, demonstrating high effectiveness for the generalist model (Table 5). In contrast, **FakeShield merely mixes all training data, without any special design for a generalist model**.

**4. Performance Gap**. Due to the above reasons, our Omni-IML has significantly lower performance degradation in joint training, and achieves significantly higher performance than FakeShield (e.g. 16.4 higher average IoU on natural IML and 24.9 higher average IoU on all tasks, Table 2, 4, 6).

**5. Structural Flexibility**. The localization module and interpretation module in our Omni-IML are relatively independent. They are trained totally independently and are connected only with a mask prediction during inference. Consequently, in practice, one can change the weights of the localization module to quickly adapt to different scenarios. In contrast, the localization and interpretation functions in FakeShield are highly coupled, limiting its flexibility.

**6. Benchmark Difference**. As shown in Table 1:

- The MMTD-Set in FakeShield paper contains only natural and face images, and only natural images have pixel-level annotations. In contrast, our Omni-273k benchmark contains natural, document, face, scene text images, and all have pixel-level annotations.

- The MMTD-Set contains only single-target scenario, while ours contains both single-target and multi-target scenarios.

- Our Omni-273k contains $10\times$ more forgeries and $20\times$ more manual (real-world, high-quality) forgeries than the MMTD-Set, making it more qualified in evaluating real-world model performance.

- Our Omni-273k adopts structured annotation format to enable reasonable evaluation and in-depth item-wise analyses. In contrast, the MMTD-Set annotation is unstructured. Only one metric can be used per evaluation, no item-wise can be produced, and thus cannot provide enough insights behind the results.

## C    MORE MODEL DETAILS

### C.1    MODAL GATE ENCODER

**Image Encoding**. Given an input image $X \in \mathbb{R}^{H \times W \times 3}$ and its Y-channel quantization table $QT \in \mathbb{R}^{8 \times 8}$, we extract vision features $F_{rgb}$ using Vision Perception Head (VPH), $F_{rgb} = VPH(X)$. We obtain frequency features $F_{freq}$ from the DCT coefficients and quantization tables (QT) of the images using Frequency Perception Head (FPH), $F_{freq} = FPH(DCT(X), QT)$. We use the same VPH and FPH architectures as those proposed in Document Tampering Detector Qu et al. (2023). The $F_{freq}$ is fused with $F_{rgb}$ by a conv-layer $Conv$ to get the fused features $F_{fused}$, $F_{fused} = Conv(F_{rgb}, F_{freq})$. Two coarse binary mask predictions $P_{rgb}$ and $P_{fused}$ are further obtained from $F_{rgb}$ and $F_{fused}$ with two auxiliary heads $AuxHead$ respectively, $P_{rgb} = AuxHead_1(F_{rgb})$, $P_{fused} = AuxHead_2(F_{fused})$, each of the auxiliary heads consists of two conv-layers.

**Modal Gate**. The input of the proposed Modal Gate has four parts: $F_{rgb}$, $F_{fused}$, $P_{rgb}$ and $P_{fused}$; We repeat $P_{rgb}$, $P_{fused}$ and concatenate them with $F_{rgb}$, $F_{fused}$ to get $F_{cat}$, which is then fed into a binary classifier for optimal modality prediction. $P_{cls} = CLS(F_{cat})$, $P_{modal} = Round(\sigma(P_{cls}))$, where $\sigma$ is the sigmoid function and $Round$ function rounds up to the nearest integer, which means a fixed threshold of 0.5 for our modal gate's binary classification. The classifier $CLS$ consists of several conv-layers, a global average pooling layer and a linear layer, and is used to determine whether to use the fused feature $F_{fused}$ or the pure vision feature as the encoder input $F_{rgb}$, by observing the noise level and anomaly significance level of $F_{fused}$, $F_{rgb}$ and their coarse predictions $P_{rgb}$ and $P_{fused}$.

**Loss Function.** The Modal Gate Encoder is optimized with $L_{MG}$, the sum of two segmentation losses and one classification loss. $CE$ denotes the cross-entropy loss function, $L_m$ is the ground-truth mask indicating tampered region and $L_c \in \{0, 1\}$ is the classification label indicating the optimal modality. $L_c$ is obtained by choosing the most accurate coarse prediction. $IoU(x, y)$ denotes the Intersection over Union between inputs $x$ and $y$.

$$L_{MG} = CE(P_{rgb}, L_m) + CE(P_{fused}, L_m) + CE(P_{cls}, L_c)$$

$$L_c = \begin{cases} 1 & IoU(P_{rgb}, L_m) > (IoU(P_{fused}, L_m) + 0.1) \\ 0 & otherwise \end{cases}$$

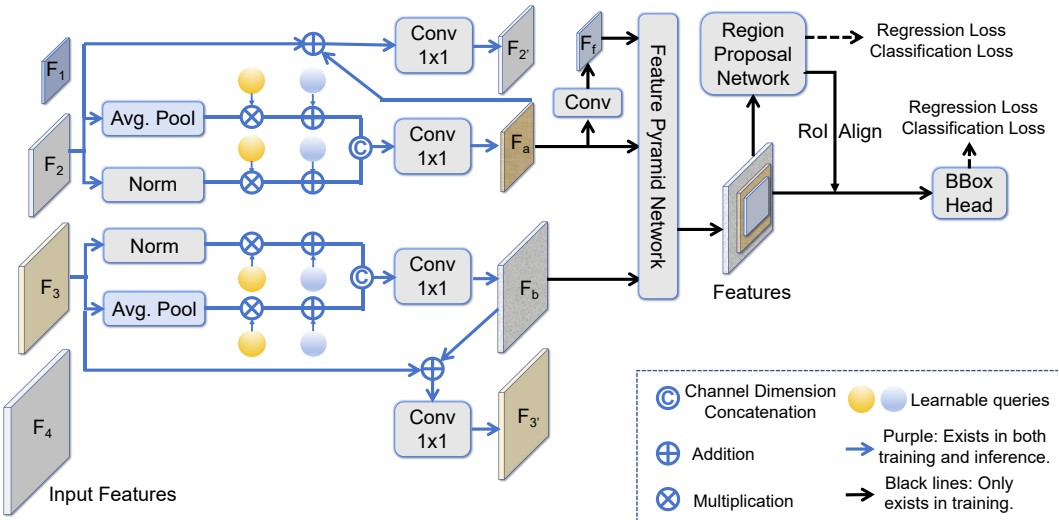

Figure 6: The proposed Anomaly Enhancement.

## C.2 ANOMALY ENHANCEMENT

**Key Idea.** Different image types from different IML tasks produce different features, joint training brings much more noise to the features and confuses the IML model. To tackle this, we propose to enhance the contrast of forged regions and improve the feature extraction across diverse image types through including an extra box supervision during training. However, directly training the model with the both detection and segmentation frameworks may also cause task competition for model parameters Heuer et al. (2021) and weaken model performance, while directly scaling up the model parameters could alleviate the competition but will increase computation burden. To address this, we propose a novel collaboration module Anomaly Enhancement (AE).

**Method**. As shown in the Figure 6, for the input features $F_2$ and $F_3$ in $\frac{1}{8}$ and $\frac{1}{16}$ input image size, we first extract task-agnostic features $F_a$ and $F_b$ with two attention layers that are used to decouple and to minimize negative impact from the segmentation supervision. After that, $F_a$ and $F_b$ are processed by the detection modules, including two Feature Pyramid Networks (FPNs) Lin et al. (2017) and the Faster R-CNN's Ren et al. (2015) Region Proposal Network (RPN) and box head. The detection modules (black arrows in the Figure 6) are only present during training. Including the two cascaded FPNs reduces parameter competition from the detection framework and discarding them during inference ensures the computation efficiency, successfully addressing the dilemma. The AE module is trained in an end-to-end manner with the same four loss functions as the Faster R-CNN, including the two classification losses and two regression losses for RPN and box head respectively. After training, the $F_a$ and $F_b$ contain positive features enhanced by the detection supervision, we add them to the original features $F_2$ and $F_3$ and fuse them with conv-layer to get $F_{2'}$ and $F_{3'}$.

The proposed AE effectively achieves task collaboration while keeping the inference cost almost unchanged. With the AE module, the tampered regions in features $F_2$ and $F_3$ can be enhanced and the false-positive noise can be reduced. Consequently, our AE module helps to extract better common features and thus benefits the generalist model.

**Loss Function**. The AE module is optimized by bounding box losses as Faster R-CNN Ren et al. (2015) from the RPN and RoI-Head. $L_{AE} = L_{cls}^{RPN} + L_{regression}^{RPN} + L_{cls}^{RoIHead} + L_{regression}^{RoIHead}$. The ground-truth boxes are the bounding boxes of the mask annotations' connected regions.

## C.3 DYNAMIC WEIGHT DECODER

In the proposed Dynamic Weight Decoder, the low-level input features are fused with high-level input features by Pyramid Pooling Module Zhao et al. (2017) and Feature Pyramid Network Lin et al. (2017) to obtain multi scale features $F_1, F_2, F_3, F_4$. A global feature vector $V_g$ is obtained by average pooling $F_1$. The extracted feature is concatenated to the multi-scale features, $F_{cat} =$

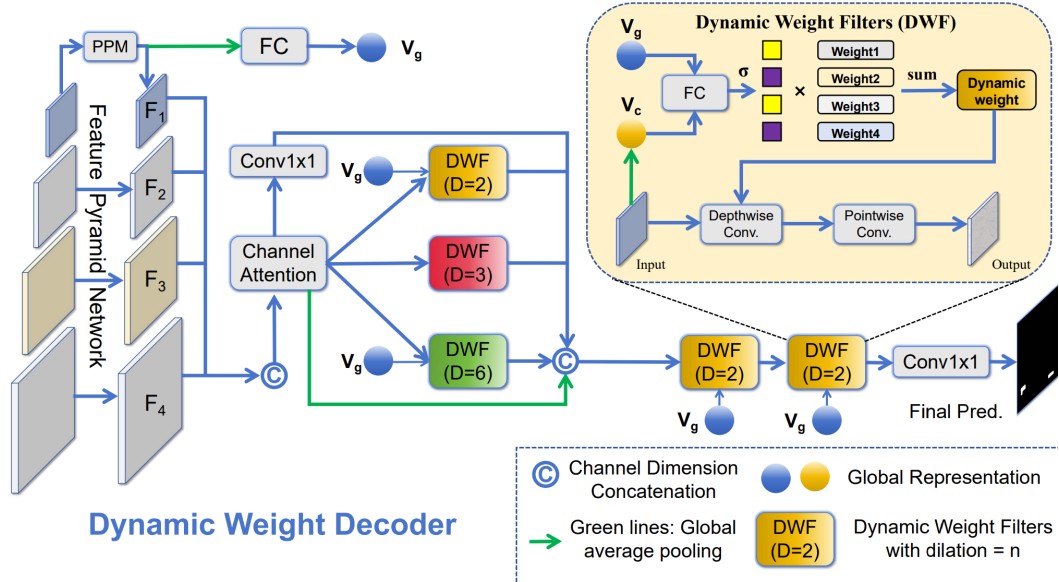

Figure 7: The proposed Dynamic Weight Decoder.

$Concat(F_1, F_2, F_3, F_4)$. The concatenated features are channel dimension reduced and processed by a series of Dynamic Weight Filters with different dilation rates,

$$F_{dec1} = Concat(Avg(F_{cat}), F_{dw}, F_{cat}),$$

$$F_{dw} = Concat([DWF_n(F_{cat}, V_g) \; for \; n \; in \; (2, 3, 6)]),$$

$DWF_n$ denotes the proposed DWF with dilation rate $n$. The final prediction $P_{DWD}$ is obtained by

$$P_{DWD} = Conv(DWD_2(DWD_2(Conv(F_{dec1}), V_g), V_g)),$$ where $Conv$ denotes $1 \times 1$ conv-layer. The DWD is supervised by minimizing the cross-entropy loss between $P_{DWD}$ and the ground-truth mask $L_m$. $L_{DWD} = CE(P_{DWD}, L_m)$

**Dynamic Weight Filters.** As shown in the top-right of Figure 7, to obtain the dynamic filters, we first average pool the input feature to obtain a current global representation $V_c$ (orange box in Figure 7), then interact $V_c$ with the global image vector $V_g$ (blue box in Figure 7) with a fully connected layer and identify the optimal dynamic filters $D_{opt}$ by weighted summation of four common convolutional filters. $A_i = \sigma(FC(V_c, V_g))$, $D_{opt} = \sum_{i=1}^{4} A_i * Wi$, $\sigma$ is the sigmoid function, $FC$ is a linear layer, $W_i$ is the $i$th filter in the Dynamic Weight Filters (DWF). Finally, we depthwise convolve the input feature with $D_{opt}$ and then perform point-wise convolution with $1 \times 1$ conv-layer to obtain the final output.

# D  DETAILED CHAIN-OF-THOUGHT PIPELINE

## D.1  DETAILED CHAIN-OF-THOUGHT

**Step 1. Instance-wise Tampered Object Recognition.** In this step, we accurately recognize the content and position for each tampered region. Given the binary mask annotation $I_{mask}$ indicating the tampered region for a tampered image $I_{tamp}$, each of the connected components in $I_{mask}$ is a tampered instance.

**Content recognition and absolute position calculation for text images**, For text images, the OCR result for each tampered instance is the content. We cut off the tampered regions of the image $I_{tamp}$ and recognize the text respectively using commercial OCR engine. The absolute position is the position relative to the entire image, and **is the one selected from the nine options: Top left, Top center, Top right, Left side, Center, Right side, Bottom left, Bottom center, Bottom right**. We calculate the absolute position for each tampered region based on the position of its centroid.

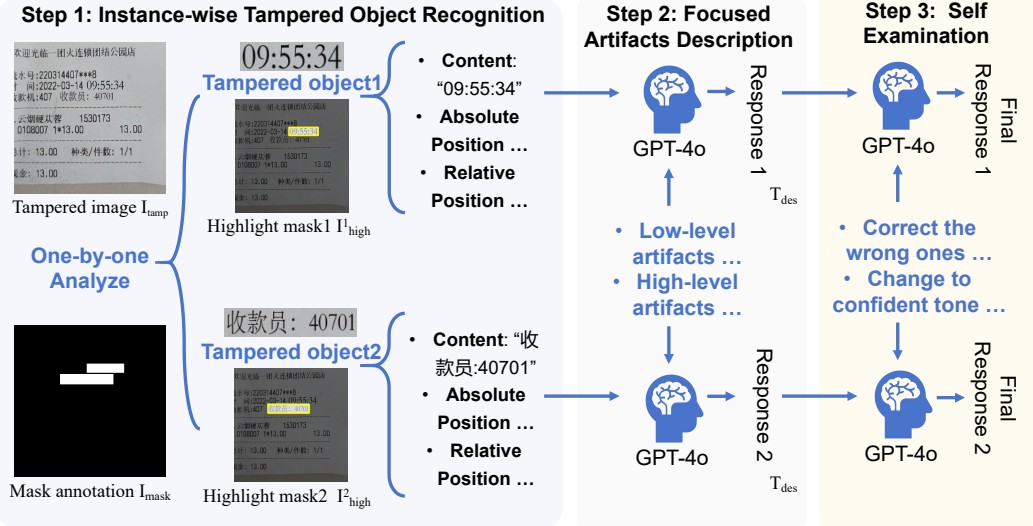

Figure 8: The proposed Chain-of-Thoughts Pipeline.

**Content recognition and absolute position calculation for non-text images**, For face and natural images, we highlight the $m$ tampered regions in $I_{tamp}$ respectively to get $m$ highlight masks $I_{high}^n$, $n \in [1, m]$. We prompt GPT-4o with $I_{tamp}$ and $I_{high}^n$ sequentially for each tampered instance to get its content (e.g. "The face of a smiling young man with black hair and in blue shirts"). We also calculate the absolute position for each tampered region based on the position of its centroid. For natural images in NIST16 Guan et al. (2019), IMD20 Novozamsky et al. (2020) and MIML Qu et al. (2024) datasets where different tampered regions can belong to the same tampered object, instead of processing each region respectively, we directly highlight all the $n$ tampered regions in the same image, and query GPT-4o to sequentially generate the content and absolute position descriptions for all the objects.

**Relative position for text images**. The relative position is the text line containing the tampered text region. We detect and recognize the text lines in the given image $I_{tamp}$, and determine which text line a tampered text region belongs based on IoU.

**Relative position for non-text images**. We query GPT-4o to generate relative position descriptions for non-text images (e.g., "Over the brown squirrel eating a nut").

**Step 2. Focused Artifact Description.** In this step, we aim to obtain detailed descriptions of image artifacts for each tampered instance. For each tampered instance, we prompt GPT-4o with $I_{tamp}$, $I_{high}^n$, the previously recognized content and position, and an elaborate query (listing the common artifact perspectives) to obtain the detailed descriptions $T_{des}$ for low-level visual and high-level semantic artifacts. The full query (taking document image an example) is shown in Figure 13.

**Step 3. Self Examination.** In challenging samples where image artifacts are not obvious, the response of GPT-4o can be unconfident and incorrect, while manual filtering is costly. To this end, we propose to further improve the annotation quality by guiding GPT-4o to carefully examine and correct its previous response. For each tampered object, we prompt GPT-4o with $T_{des}$, $I_{tamp}$, $I_{high}^n$, and the previously recognized contents and positions, then show it an example containing both an unconfident, incorrect answer and the corresponding manually corrected one. The full query (taking document image an example) is shown in Figure 14.

### D.2 QUALITY EVALUATION

To evaluate the performance of the proposed annotation pipeline. We have volunteers score 500 random samples from each of the four IML image types, with 0 being the lowest quality and 5 being the highest. The average score for the annotations generated with our method is 4.87, which is higher than the 2.94 for annotations generated without our method. This validates that our pro-

Table 7: Statistics of the Omni-273k dataset. "Image num." denotes the number of tampered images, 'Target num.' denotes the average number of tampered instances in each sample. 'Avg. len.' denotes the average length of the annotation chars in each sample.

| Image Domain | Image num. | Target num. | Avg. len. |
|---|---|---|---|
| Natural images | 126366 | 1.269 | 1925 |
| Document images | 80028 | 1.481 | 2131 |
| Face images | 60689 | 2.078 | 1265 |
| Scene text images | 1978 | 2.533 | 3379 |

posed pipeline can significantly improve annotation quality and produce close-to-human annotation quality.

# E   More Details about the Omni-273k Dataset

**Statistics**. The statistics of our dataset are shown in Table 7. A representative sample displaying our Omni-273k's structured annotation format is illustrated in Figure 18.

**Train/Test Split**. The Omni-273k dataset is constructed by adding forgery interpretation annotations to existing high-quality datasets. Therefore, the split of training and test sets of Omni-273k follows the conventional split in each image domain. To be specific, 217, 907 images from CASIAv2 Dong et al. (2013), MIML Qu et al. (2024), the training sets of DocTamper Qu et al. (2023), SACP Alibaba Security (2020), OpenForensics Le et al. (2021), Tampered-IC13 Wang et al. (2022b) and OSTF Qu et al. (2025b) are split as the training set, the rest 55, 869 images are split as the test set.

**Manual Cleaning**. We manually reviewed and cleaned the forgery interpretation annotations of the Omni-273k test set. The textual annotations of the training set are the output of our CoT annotation pipeline and may contain minimal noise. However, this minimal noise better simulates the noisy real-world scenarios and reflects model robustness.

**Effectiveness**. Our Omni-273k can help large language models to better interpret image artifacts. The Omni-IML model trained with our Omni-273k can effectively describe the tampered region content, the absolute position, relative position and artifact clues of the forgeries from all the four major image domains (natural image, document, scene text, face), as shown in Figures 15, 16, 17.

# F   More Implementation Details

The localization module and the interpretation module of our Omni-IML are trained totally independently, **without any effect or involvement to each other in training**.

## F.1   Image Forgery Localization

**Model Training Details**. The backbone model of our Omni-IML is ConvNeXt-Base Liu et al. (2022b) initialized with its official ADE20k Zhou et al. (2017) pre-trained weights, following previous works Yu et al. (2024); Qu et al. (2024). The Omni-IML is trained with the cross-entropy loss for 400k iterations, using the AdamW optimizer Loshchilov & Hutter (2017), with a batch size of 16 and an input size of $512 \times 512$. The initial learning rate is set to 1e-4 and decays to 1e-6 in a linear schedule. We remove the forgery interpretation module when training the forgery localization module. A fixed threshold of 0.5 is used to binarize model predictions during inference. Pixel-level Intersection over Union (IoU) and binary F1 are used to evaluate model performance.

**More Details about the Training Data**. The natural image part of our training data includes: tamperCOCO Kwon et al. (2022), CASIAv2 Dong et al. (2013), MIML Qu et al. (2024) and COCO Lin et al. (2014). These datasets have 600,000, 5123, 123150 and 118287 images respectively, and the sampling ratio for these datasets is approximately 2:2:2:1. The document image part of our training data includes: SACP Alibaba Security (2020) training set, DocTamper Qu et al. (2023) training set. These datasets have 1604 and 120,000 images respectively, the sampling ratio is approximately 1:2. The face image part of our training data is the training set of OpenForensics Le et al. (2021), which has 44122 images. The scene text image part of our training data includes the training set of Tampered-IC 13 Wang et al. (2022b), which has 229 images. The total number of the training data is 1012515, the sum of the above numbers. The sampling ratio for natural images, documents, face images, scene text images is approximately 10: 5: 2: 20.

**More Details about the Test Data**. The test data for pixel-level IML includes CASIAv1 Dong et al. (2013) with 921 images, Coverage Wen et al. (2016) with 100 images, NIST16 Guan et al. (2019) with 582 images, CocoGlide Guillaro et al. (2023) with 512 images, IMD20 Novozamsky et al. (2020) with 2,010 images, SACP Alibaba Security (2020) test set with 406 images, DocTamper Qu et al. (2023) test set with 30,000 images and minq=75 ('minq' is the minimum compression quality factor), DocTamper-FCD Qu et al. (2023) with 2,000 images and minq=75, DocTamper-SCD Qu et al. (2023) with 18,000 images and minq=75, OpenForensics Guo et al. (2023) test set with 18,895 images, Tampered-IC13 Wang et al. (2022b) test set with 233 images and OSTF Qu et al. (2025b) test set with 1007 images.

**Evaluation Metrics**. For the DocTamper benchmark, we use its official scripts to evaluate model performance. For other benchmarks, we calculate foreground IoU and pixel-level Precision (P), Recall (R), and F1-score (F) for each sample and then compute the average score following previous work Ma et al. (2024) for fair comparison.

**Compared Methods**. In Table 2, the compared methods include RRU-Net Bi et al. (2019), MVSS-Net Dong et al. (2022), PSCC-Net Liu et al. (2022a), CAT-Net Kwon et al. (2022), IF-OSN Wu et al. (2022), EVP Liu et al. (2023b), TruFor Guillaro et al. (2023), APSC-Net Qu et al. (2024), Sparse-ViT Su et al. (2025) and PIM Kong et al. (2025) for natural IML; CFL-Net Niloy et al. (2023), TIFDM Dong et al. (2024), Swin-UPer Liu et al. (2021), DTD Qu et al. (2023), CAFTB Song et al. (2025) for Document IML; MantraNet Wu et al. (2019), HPFCN Li & Huang (2019), DOA-GAN Islam et al. (2020), HiFi-Net Guo et al. (2023), MoNFAP Miao et al. (2024) for face IML; DeepLab3 Chen et al. (2018), HRNetv2 Wang et al. (2020), BEiT-UPer Bao et al. (2021), Seg-Former Xie et al. (2021), UPOCR Peng et al. (2024) for scene text IML.

## F.2 IMAGE FORGERY INTERPRETATION

**Hyper-parameters**. The multimodal large language model in our interpretation module is LoRA Hu et al. (2022) fine-tuned 60k iterations with a batch-size of 16, a LoRA rank of 64, an AdamW optimizer with learning rate decayed from 1e-4 to 0. We fine-tune our forgery interpretation module after the training of localization model is done. During both the fine-tuning and inference stages of our forgery interpretation module, the input masks are the predictions from the trained forgery localization module.

**LLM Query Details**. For the interpretation model fine-tuned with our method, the query prompt is "The left half of the input image is a tampered face image. The right half of the input image is a reference image with the suspected tampered region of the left half image highlighted, and the suspected authentic region darkened. Please find the tampered region in the left half of the input image. For each tampered region, please describe its content, absolute position, relative position, and visible artifacts. The description of the visible artifacts should be in JSON format, where the keys are the types of visible artifacts and the values are the corresponding detailed analysis. Always assume that you are observing only the left half of the input image".

For the interpretation model fine-tuned without our method, the query prompt is "The input image is a tampered image. Please find the tampered region in the left half of the input image. For each tampered region, please describe its content, absolute position, relative position, and visible artifacts. The description of the visible artifacts should be in JSON format, where the keys are the types of visible artifacts and the values are the corresponding detailed analysis. Always assume that you are observing only the left half of the input image".

For the pre-trained model on zero-shot testing, the query prompt is "The input image is a tampered image. Please describe the content, absolute position, relative position, and image artifacts of the tampered object(s) in this image. Content is the appearance of the tampered object (e.g. "A white cat sleeping on the table") or OCR of the tampered text. Absolute position is the position relative to the entire image (e.g., "Top left of the image"). Relative position is the position relative to other objects (e.g. "In front of the women in white shirts") or the text line of the tampered text (e.g. "In the text line: 09:34:50"). Image artifacts can include edge artifacts, texture artifacts, font artifacts, alignment artifacts, lighting anomaly, depth anomaly, color inconsistency, semantic artifacts, etc. If there are multiple tampered objects in this image, please describe the content, absolute position, relative position, and image artifacts for each of them".

Table 8: Ablation study on whether a soft or hard modality switch is better in the Modal Gate. We used the hard one.

| Ablation of MG | Natural | | Document | | Scene Text | | Face | | Average | |
|---|---|---|---|---|---|---|---|---|---|---|
| | IoU | F1 | IoU | F1 | IoU | F1 | IoU | F1 | IoU | F1 |
| Soft | .479 | .600 | .628 | .731 | .536 | .696 | .878 | .925 | .630 | .738 |
| Hard | **.612** | **.678** | **.766** | **.813** | **.610** | **.749** | **.923** | **.957** | **.728** | **.799** |

Table 9: Ablation study on the Modal Gate's another input modality besides pure vision.

| Input Modality | Natural | | Document | | Scene Text | | Face | | Average | |
|---|---|---|---|---|---|---|---|---|---|---|
| | IoU | F1 | IoU | F1 | IoU | F1 | IoU | F1 | IoU | F1 |
| None | .538 | .625 | .634 | .742 | .561 | .712 | .904 | .947 | .659 | .756 |
| SRM | .545 | .629 | .637 | .743 | .578 | .723 | .916 | .953 | .669 | .762 |
| Bayar | .498 | .599 | .603 | .722 | .544 | .700 | .906 | .948 | .638 | .742 |
| NP++ | .540 | .626 | .611 | .727 | .560 | .712 | **.925** | **.958** | .659 | .756 |
| DCT | **.612** | **.678** | **.766** | **.813** | **.610** | **.749** | .923 | .957 | **.728** | **.799** |

Table 10: Modal Gate selection rate of pure vision features and frequency+vision fused features under four IML tasks.

| Features | Natural IML | Document IML | Scene Text IML | Face IML |
|---|---|---|---|---|
| Pure vision | 0.94 | 0.02 | 0.78 | 0.97 |
| Fused | 0.06 | 0.98 | 0.22 | 0.03 |

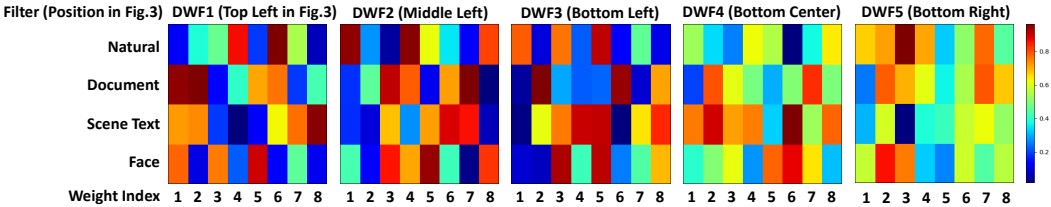

Figure 9: Dynamic Weight Filters have different weight activation values for different tasks. Each cell is a filter weight activation value.

# G MORE EXPERIMENTAL RESULTS

## G.1 MORE ABLATION STUDIES ON THE MODAL GATE

**Soft or Hard Gate**. In our Modal Gate, a soft gate means that we fuse pure vision features $F_{rgb}$ and vision+frequency features $F_{fused}$ through adaptive weighting with a channel-spatial attentional module. A hard gate means that we use $F_{rgb}$ or $F_{fused}$. The results in Table 8 show that the hard gate we used is better. This is because the motivation of our Modal Gate is to thoroughly exclude the distortion from the frequency features when the frequency features are too noisy. Our hard gate can achieve this, whereas the soft gate cannot.

**Input Modality**. We evaluated the performance of our model under different input noise domains of our modal gate encoder, including SRM Zhou et al. (2018), Bayar Dong et al. (2022), DCT Qu et al. (2023), and NoisePrint++ (NP++) Guillaro et al. (2023). As shown in Table 9, the DCT auxiliary modality we adopted is the best.

**Modal Gate Activations**. Table 10 shows the selection ratio of RGB/fused features in test data from different image domains. Frequency+vision fused features are typically used in text images and pure vision features are mostly used in natural and face images.

## G.2 DWD FILTER ACTIVATIONS

Figure 9 shows the filter weight activations of the dynamic weight filters in the Dynamic Weight Decoder (DWD). These values are calculated by averaging the activations of each test sample in each task. The results demonstrate that the filters have different activation weights for different tasks. The filter names correspond to Figure 7.

Table 11: Experiments on the interpretation ability. 'Cos. ' denotes the cosine similarity of the paragraph vectors. 'Avg.' denotes average score.

| Method | Cos. | ROUGE-2 | ROUGE-L | BLEU | Avg. |
|---|---|---|---|---|---|
| Pre-trained models (**zero-shot**). | | | | | |
| GPT-4o | .975 | .112 | .201 | .074 | .341 |
| InternVL3 2B Chen et al. (2024a) | .973 | .063 | .135 | .017 | .297 |
| Qwen2.5-VL 3B Bai et al. (2025) | .964 | .077 | .151 | .020 | .303 |
| Qwen2.5-VL 7B Bai et al. (2025) | .972 | .094 | .175 | .035 | .319 |
| Supervise Fine-Tuned (**SFT**) models. | | | | | |
| DeepSeek-VL 7B Lu et al. (2024) | .993 | .385 | .441 | .308 | .532 |
| MiniCPM-V-2.6 8B Yao et al. (2024) | .992 | .378 | .439 | .287 | .524 |
| InternVL3 2B Chen et al. (2024a) | .993 | .372 | .432 | .293 | .523 |
| Qwen2.5-VL 3B Bai et al. (2025) | .994 | .403 | .456 | .316 | .542 |
| Qwen2.5-VL 7B Bai et al. (2025) | .994 | .403 | .455 | .316 | .542 |
| Supervise Fine-Tuned (SFT) models **with our method**. | | | | | |
| Ours+InternVL3 2B | .993 | .405 | .466 | .340 | .551 |
| Ours+Qwen2.5-VL 3B | .995 | .424 | .489 | .363 | .568 |
| Ours+Qwen2.5-VL 7B | **.995** | **.433** | **.498** | **.367** | **.573** |

## G.3 MORE FORGERY INTERPRETATION RESULTS

In Table 11, we directly evaluate models with their whole output string for each sample using word-vectors cosine similarity Mikolov et al. (2017), ROUGE and BLEU, following the previous works Xu et al. (2024); Huang et al. (2024). The compared MLLMs include GPT-4o, DeepSeek-VL-7B Lu et al. (2024), MiniCPMV2.6-8B Yao et al. (2024), InternVL3-2B Chen et al. (2024a) and InternVL2-8B, Qwen2.5-VL-3B Bai et al. (2025) and Qwen2.5-VL-7B. All the pre-trained models (the top group in Table 11) show poor interpretation performance (e.g. all BLEU scores less than 0.1), confirming that the pre-trained LLMs are not naturally adequate to detect and explain image forgery. Fine-tuning the models with our dataset is necessary for them to achieve forgery interpretation capability. The models fine-tuned with our method (bottom group) consistently outperform those fine-tuned without our method (middle group) under all metrics, confirming the effectiveness of the reference visual prompt in our interpretation module. We have presented the coarse-grained evaluation of our model in Table 11 of our paper, which further validates the conclusion.

## G.4 ROBUSTNESS EVALUATION

**Forgery Localization**. We test the robustness of our localization model by applying image distortion to NIST16 dataset images, following the standard setting in the field of image manipulation localization Li et al. (2023); Zhou et al. (2023); Li et al. (2024b). We compare the pixel-level AUC performance with other methods, including Mantra-Net Wu et al. (2019), MVSS-Net Dong et al. (2022), SPAN Hu et al. (2020), PSCC-Net Liu et al. (2022a), ObjectFormer Wang et al. (2022a), SAFL-Net Sun et al. (2023), NCL Zhou et al. (2023), ERMPC Li et al. (2023), UnionFormer Li et al. (2024b). Table 12 shows that our method exhibits strong robustness to various distortion operations, significantly outperforming previous works.

**Forgery Interpretation Robustness**. We test the robustness of our localization model by applying various image distortions to the test set. The results are shown in Table 13, which have confirmed the robustness of our method.

## G.5 EVALUATION ON UNKNOWN IML TASKS

Our method is designed to alleviate performance degradation in multi-task joint training, rather than generalize to unseen tasks. Nevertheless, we evaluated the joint-task-trained models' performance on an unseen IML task, the remote sensing IML Zhang et al. (2024), with its RSCMQA dataset. The results are shown in Table 14. Our method achieves state-of-the-art performance and demonstrates great potential for generalization to unseen IML tasks.

Table 12: Manipulation localization performance on NIST16 dataset under various distortions. AUC scores are reported, 'Ori' represents no distortion. 'k' represents kernel size of the Gaussian blur and 'q' represents quality in JPEG compression.

| Method | Ori | Resize | | Blur | | JPEG | |
|---|---|---|---|---|---|---|---|
| | | .78x | .25x | k=3 | k=15 | q=100 | q=50 |
| ManTra-Net Wu et al. (2019) | .795 | .774 | .755 | .774 | .746 | .779 | .744 |
| MVSS-Net Dong et al. (2022) | .788 | .783 | .775 | .786 | .758 | .788 | .788 |
| SPAN Hu et al. (2020) | .840 | .832 | .802 | .831 | .792 | .836 | .807 |
| PSCC-Net Liu et al. (2022a) | .855 | .853 | .850 | .854 | .800 | .854 | .854 |
| ObjectFormer Wang et al. (2022a) | .872 | .872 | .863 | .860 | .803 | .864 | .862 |
| SparseViT Su et al. (2025) | .888 | .884 | .869 | .881 | .877 | .886 | .881 |
| NCL Zhou et al. (2023) | .912 | .856 | .831 | .840 | .806 | .843 | .819 |
| ERMPC Li et al. (2023) | .895 | .893 | .877 | .892 | .871 | .894 | .888 |
| UnionFormer Li et al. (2024b) | .881 | .873 | .871 | .865 | .843 | .880 | .879 |
| Ours | **.918** | **.914** | **.895** | **.892** | **.882** | **.917** | **.890** |

Table 13: Robustness evaluation for the interpretation module. Our model uses Qwen2.5-VL as its base MLLM. The score in each cell is the average cosine similarity, ROUGE-2, ROUGE-L, and BLEU scores.

| Method | Ori | Resize | | Blur | | JPEG | |
|---|---|---|---|---|---|---|---|
| | | .78x | .25x | k=3 | k=15 | q=100 | q=50 |
| DeepSeek-VL 7B Lu et al. (2024) | .532 | .529 | .510 | .527 | .514 | .528 | .517 |
| MiniCPM_V2.6 8B Yao et al. (2024) | .524 | .519 | .506 | .520 | .508 | .521 | .511 |
| InternVL3 2B Chen et al. (2024a) | .523 | .518 | .500 | .518 | .504 | .519 | .509 |
| Qwen2.5-VL 3B Bai et al. (2025) | .542 | .538 | .523 | .538 | .526 | .540 | .531 |
| Qwen2.5-VL 7B Bai et al. (2025) | .542 | .539 | .526 | .539 | .530 | .541 | .533 |
| Ours + Qwen2.5-VL 3B | .568 | .564 | .552 | .563 | .553 | .567 | .557 |
| Ours + Qwen2.5-VL 7B | **.573** | **.569** | **.559** | **.559** | **.562** | **.572** | **.565** |

Table 14: Image manipulation localization on RSCMQA dataset Zhang et al. (2024) (unknown remote sensing IML task). Pixel-level binary F1 metric is used. 'Baseline' is our baseline model.

| Method | HiFi-Net | MVSS-Net | CAT-Net | TruFor | APSC-Net | FakeShield | SparseViT | DTD | Baseline | Ours |
|---|---|---|---|---|---|---|---|---|---|---|
| F1 | .103 | .133 | .212 | .216 | .258 | .220 | .158 | .147 | .209 | **.399** |

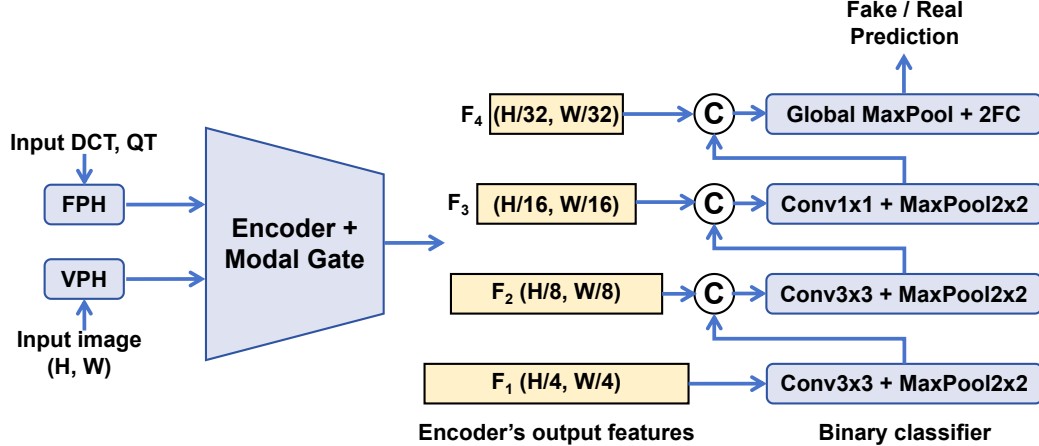

Figure 10: The classifier for image-level forgery detection.

## G.6 Image-level Forgery Detection

The image-level forgery detection is achieved by a binary classifier, as shown in Figure 10. The classifier extracts features from the encoder's output features, with alternating conv-layers and max pooling layers. The classifier performs the final classification with global max pooling and two fully connected layers. Binary cross entropy loss is used to optimize the classifier. We test the image-level forgery detection performance of our model with AUC and balanced accuracy, following the same standard setting in TruFor Guillaro et al. (2023). The results in Table 15 show that the proposed model is also very good at image-level classification and significantly outperforms previous works.

Table 15: Comparison study on image-level forgery detection. 'Acc' represents balanced accuracy. 'mAUC' represents mean AUC, 'mAcc' represents mean balanced accuracy.

| Method | CASIA1 | | Columbia | | Coverage | | CocoGlide | | Average | |
| --- | --- | --- | --- | --- | --- | --- | --- | --- | --- | --- |
| | AUC | Acc | AUC | Acc | AUC | Acc | AUC | Acc | mAUC | mAcc |
| CR-CNN Liu & Zhao (2020) | .670 | .535 | .755 | .628 | .553 | .510 | .589 | .533 | .642 | .552 |
| RRU-Net Bi et al. (2019) | .574 | .488 | .583 | .500 | .482 | .500 | .533 | .503 | .543 | .498 |
| ManTraNet Wu et al. (2019) | .644 | .500 | .810 | .500 | .760 | .500 | .778 | .500 | .748 | .500 |
| SPAN Hu et al. (2020) | .480 | .487 | .999 | .951 | .670 | .605 | .475 | .491 | .656 | .634 |
| CAT-Net Kwon et al. (2022) | .942 | .838 | .977 | .803 | .680 | .635 | .667 | .580 | .817 | .714 |
| IF-OSN Wu et al. (2022) | .735 | .635 | .882 | .522 | .557 | .510 | .611 | .567 | .696 | .559 |
| MVSS-Net Dong et al. (2022) | .932 | .808 | .984 | .667 | .733 | .545 | .654 | .536 | .826 | .639 |
| PSCC-Net Liu et al. (2022a) | .869 | .683 | .300 | .508 | .657 | .550 | .777 | .661 | .651 | .601 |
| TruFor Guillaro et al. (2023) | .916 | .813 | **.996** | **.984** | .770 | .680 | .752 | .639 | .859 | .779 |
| Omni-IML (Ours) | **.950** | **.891** | **.996** | .967 | **.800** | **.750** | **.890** | **.785** | **.909** | **.848** |

## H More Analyses

### H.1 Ablation for model backbone

We experimented with a lightweight backbone. By replacing ConvNeXt-Base with ConvNeXt-Small, we created a model that is significantly smaller and faster while achieving comparable performance (Table 17), demonstrating the improved trade-off.

### H.2 Feedback from explanation to localization for error correction

We used the model's textual output (object content and position) to guide a second-stage refinement. Specifically, we fed the explanations for natural/face images into an open-vocabulary segmentation model (DINO-X Ren et al. (2025)) and the explanations for document/scene-text images into an OCR tool (OCRSpace from GitHub) to generate refined localization masks. The results in Table 18, show a consistent improvement in localization accuracy. This confirms that feedback from explanation can refine localization.

### H.3 Ablation study of the cross-domain sampling ratio

The sampling ratio represents how many times images from each domain are repeated when constructing the online training dataset. During training, each batch is sampled randomly from this dataset. Our sampling ratio is empirically derived from two principles: task difficulty and image resolution. First, we assign a higher sampling weight to more challenging domains (e.g., scene-text). Second, we account for resolution, as larger images (like those in the scene-text domain) yield more potential training crops. This principled approach, which directly addresses the small dataset size by considering available training area, is validated by our ablation study. The results in Table 19 confirm our ratio is near-optimal and effectively balances against overfitting and underfitting.

### H.4 Comparison with non-CoT pipeline in Anomaly Description Annotation

For a fair quantitative comparison, we randomly selected 400 test samples from each domain and generated tampered-region explanations using the previous non-CoT approach (as in FakeShield Xu

et al. (2024)). We then compared these non-CoT outputs with our recorded CoT responses, evaluating both against our manually cleaned ground-truth using the same accuracy and mean-of-ROUGEL-BLEU metrics for anomaly description. The results, shown in the Table 20, clearly demonstrate the advantages of our CoT reasoning, particularly in multi-object scenarios, such as text and face images.

### H.5    ON THE NOVELTY OF THE MODAL GATE

Our Modal Gate is fundamentally different from existing fusion mechanisms in two key aspects: Binary Hard Gate: Unlike prior work using soft attention for feature fusion, our module employs a binary hard gate. This decisive mechanism proves significantly more effective, as validated by our ablation study in Table 8. Prediction-Informed Gating: Conventional gates fuse features directly. In contrast, our Modal Gate makes its decision by also considering the confidence of coarse predictions derived from each modality. This allows the gate to assess the reliability of each feature set for the specific input, leading to a more intelligent and effective fusion. The ablation study in Table 21 confirms this design choice yields superior performance.

### H.6    ON THE NOVELTY OF THE DYNAMIC WEIGHT DECODER

Our design overcomes a key limitation of standard dynamic convolutions. While conventional methods generate weights using only local features from their receptive field, our Dynamic Weight Decoder incorporates an additional global context vector ($V_g$ in Figure 7). This vector, which summarizes the entire feature map, informs the decoder about the global image type (e.g., natural vs. document). As confirmed by Table 22 this global awareness enables the generation of more specialized and effective dynamic weights, a clear departure from prior methods.

### H.7    BEHAVIOR ON MISSED DETECTIONS

The explanation module is not strictly bound by the localization output. It uses the mask as a prompt, but the MLLM can override or correct this prompt based on the full image context. We provide a qualitative example of this error-correction capability in Figure 20.

### H.8    SPLITTING THE IMAGE INTO TAMPERED OBJECTS FROM THE BINARY MASK

This is mostly achieved by connected-components analysis Bailey & Johnston (2007), as demonstrated in Figure 19.

## I    VISUALIZATION

### I.1    VISUALIZATION OF THE AE ENHANCED FEATURES

Figure 11 shows that our proposed Anomaly Enhancement module can enhance the feature contrast between forged region and authentic region.

### I.2    VISUALIZATION FOR MODULE ABLATIONS

Figure 5 shows the qualitative ablation of the modules proposed in Omni-IML. 'w.o. MG*' denotes the model without Modal Gate and using the pure vision modality; 'w.o. DWD' represents the model without the Dynamic Weight Decoder; 'w.o. DW' is the model with the DWD structure but the filter weights in the decoder keep all the same for each input; 'w.o. AE' is the model without the proposed Anomaly Enhancement module; The model without any of the proposed modules serves as the 'Baseline' model.

### I.3    QUALITATIVE COMPARISON FOR FORGERY LOCALIZATION

The qualitative comparison between RRU-Net Bi et al. (2019), PSCC-Net Liu et al. (2022a), MVSS-Net Dong et al. (2022), CAT-Net Kwon et al. (2022), TruFor Guillaro et al. (2023), APSC-Net Qu et al. (2024) is shown in Figure 12. Our proposed method demonstrates strong generalization.

Table 16: Model complexity of MVSS-Net Dong et al. (2022), SparseViT Su et al. (2025), PIM Kong et al. (2025), CAT-Net Kwon et al. (2022), APSC-Net Qu et al. (2024), DTD Qu et al. (2023), UPOCR Peng et al. (2024), FakeShield Xu et al. (2024). The Frame Per Second (FPS) is evaluated on 3090 GPU with a batchsize of 1.

| Model | MVSS-Net | SparseViT | PIM | CAT-Net | APSC-Net | DTD | UPOCR | Ours | FakeShield |
|-------|----------|-----------|-----|---------|----------|-----|-------|------|------------|
| Size | 146M | 50M | 178M | 114M | 144M | 66M | 192M | 152M | 1300M |
| FPS | 20.4 | 18.7 | 8.6 | 9.2 | 7.8 | 8.5 | 7.0 | 7.1 | 0.4 |

### I.4 QUALITATIVE COMPARISON FOR FORGERY INTERPRETATION

The model predictions for forgery interpretation are shown in Figures 15, 16, 17. The results validate that our proposed dataset and method can considerably advance the model's capability of forgery interpretation in natural language.

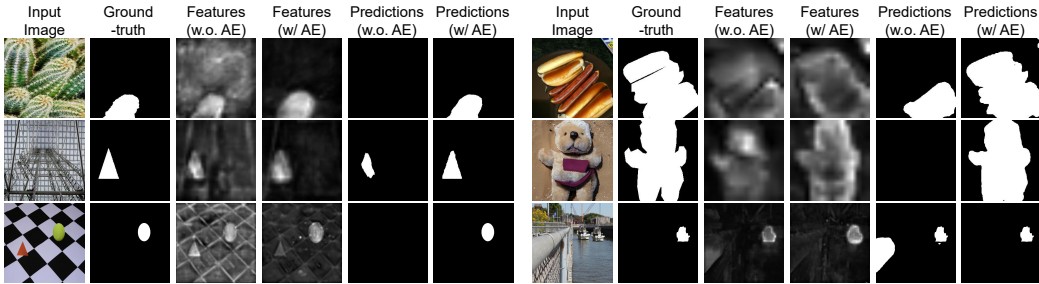

Figure 11: The proposed Anomaly Enhancement module can enhance the feature contrast between fake and real regions.

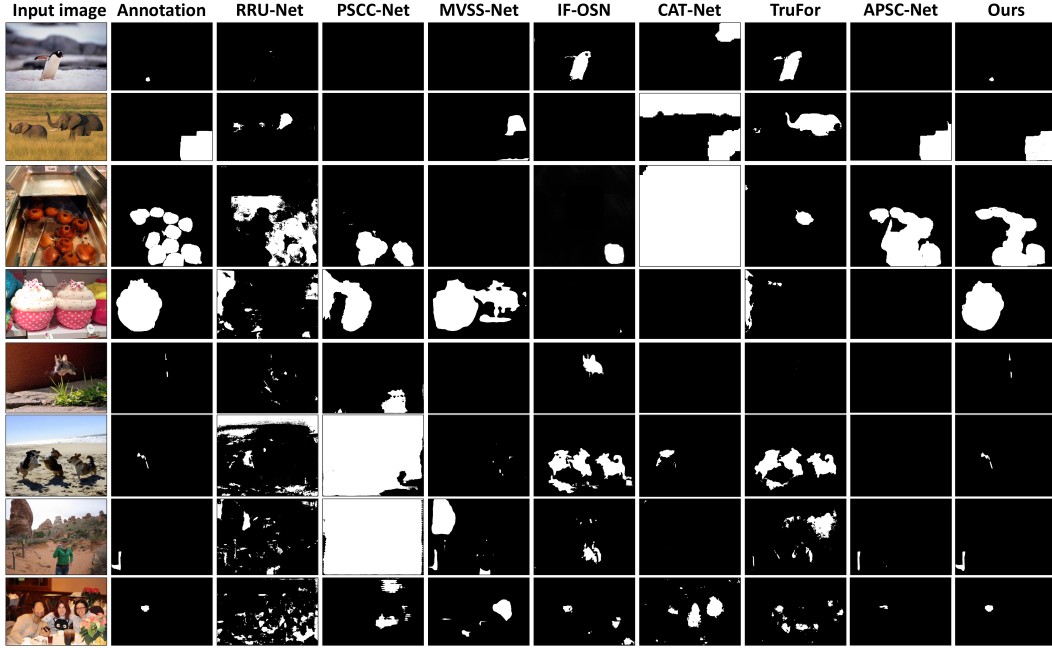

Figure 12: Qualitative comparison between different image manipulation localization models.

## J LIMITATIONS

The recent unified models in most fields are relatively large in size Wang et al. (2023), no exception to the Omni-IML. The complexity of Omni-IML's localization module is shown in Table 16. The

Table 17: Ablation study demonstrating that a lightweight backbone offers a compelling trade-off for Omni-IML, achieving comparable performance (Pixel-level IoU) with significantly reduced size and latency.

| Method | Size | FPS | Natural | Document | SceneText | Face | Average |
|---|---|---|---|---|---|---|---|
| ConvNeXt-Base | 152M | 7.1 | .612 | .766 | .610 | .923 | .728 |
| ConvNeXt-Small | 113M | 9.3 | .590 | .745 | .587 | .919 | .710 |

Table 18: Ablation study on refining localization predictions using the model's textual explanation. Performance is measured in pixel-level IoU.

| Method | Natural | Document | Scene Text | Face | Average |
|---|---|---|---|---|---|
| Original Localization | .612 | .766 | .610 | .923 | .728 |
| + Explanation Refinement | **.636** | **.774** | **.641** | **.925** | **.744** |

Table 19: Ablation study of the cross-domain sampling ratio. Performance is measured in pixel-level IoU.

| Sampling Ratio (Nat.:Doc.:Face:S.T.) | Natural | Document | Face | Scene Text | Average |
|---|---|---|---|---|---|
| 10:5:2:20 (Ours) | .612 | .766 | .923 | .610 | .728 |
| 10:5:10:20 (More face) | .603 | .754 | .925 | .596 | .719 |
| 10:5:2:50 (More scene text) | .611 | .766 | .923 | .601 | .725 |
| 10:5:2:5 (Less scene text) | .614 | .764 | .923 | .549 | .712 |

Table 20: Ablation study of the chain-of-thought (CoT) pipeline in anomaly description generation. 'Acc.' denotes accuracy. 'MRB' denotes the mean score of ROUGE-L and BLEU.

| Ablation | Doc.-Acc. | Doc.-MRB | S.T.-Acc. | S.T.-MRB | Nat.-Acc. | Nat.-MRB | Face-Acc. | Face-MRB |
|---|---|---|---|---|---|---|---|---|
| non-CoT | .673 | .815 | .521 | .797 | .802 | .870 | .695 | .924 |
| CoT (Ours) | **.949** | **.892** | **.968** | **.903** | **.966** | **.921** | **.989** | **.970** |

Table 21: Ablation study of the involvement of coarse prediction features in our Modal Gate. Performance is measured in pixel-level IoU.

| Ablation | Natural | Document | Face | Scene Text | Average |
|---|---|---|---|---|---|
| Without coarse prediction features | .560 | .742 | .602 | .915 | .705 |
| With coarse prediction features | **.612** | **.766** | **.610** | **.923** | **.728** |

Table 22: Ablation study of the global vector $V_g$ in our Dynamic Weight Decoder. Performance is measured in pixel-level IoU.

| Ablation | Natural | Document | Face | Scene Text | Average |
|---|---|---|---|---|---|
| Without global vector $V_g$ | .565 | .721 | .584 | .918 | .697 |
| With global vector $V_g$ | **.612** | **.766** | **.610** | **.923** | **.728** |

complexity of Omni-IML's interpretation module is the same as its base MLLM (e.g. Qwen2.5-VL-3B). Our single model can address four IML tasks, and its complexity is significantly lower than that of maintaining four IML models for the four IML tasks. Moreover, our model's primary strength is its performance rather than its speed. Further work will be conducted to reduce the size and improve computational efficiency.

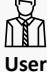

**User**

You are an expert good at analyzing tampered text images. You will be provided with the OCR information of the tampered area and two images, **the first is the tampered text image A and the second is the reference image B, with the tampered areas highlighted in the yellow box and the authentic areas darkened**. The tampered text **reads "5"**, and is located at the **top right** of the images A and B Your task is to: Describe visible details in the image that have been tampered with. Please consider the visible details caused by tampering from these perspectives. You can explain the judgments basis by combining multiple perspectives::

**1. Edge artifacts.** The background of the tampered text may be inconsistent with the authentic regions. Therefore, the edges around the tampered text region may be discontinuous and inconsistent with the background.
**2. Font inconsistency.** The font of the tampered text may be slightly different in color, size, thickness, or style from the surrounding authentic text.
**3. Alignment inconsistency.** the tampered text may have inconsistent spacing with the surrounding text or a small offset to the text line.
**4. Inconsistent alignment.** The tampered text may have inconsistent spacing with the surrounding text or a small offset to the text line.
**5. Semantic artifacts.** Tampered text may break the coherence of the sentence. For example, given a tampered text "MEED" in the sentence "I meat books.", the tampered text "meat" makes the sentence "I meat books." incoherent. The tampered text itself may appear to be misspelled.
**6. Texture anomaly.** The texture appearance of the tampered text may be different from the surrounding authentic text. For example the tampered text region may have different levels of resolution, pixelation, sharpness, or image compression artifacts.input image A.

Please use the numbers to list the points that the tampered text region has. If there are other visible anomalies exist in the tampered text region but not included in the above points, you should also describe them. Don't mention the image B or the yellow box in your answer, always assume that you are only observing the input image A.

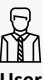

**User**

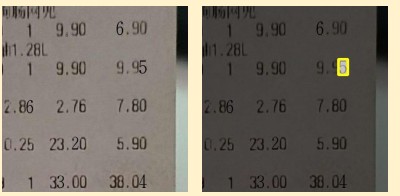

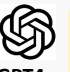

**GPT4o**

1. **Edge artifacts**: The edges of the tampered text "5" appear to have a slight blur or lack of sharpness compared to the surrounding authentic text. This inconsistency may indicate that the background does not seamlessly blend with the edges of the tampered area, leading to visible discontinuities.
2. **Font inconsistency**: The font of the tampered text "5" seems slightly different in color and thickness when compared to the surrounding numbers. This variation can draw attention to the tampered text, as it does not match the uniformity of the authentic text in size and style.
3. **Alignment inconsistency**: The placement of the tampered text "5" seems to be slightly misaligned compared to the authentic text above and below it. This offset may create an awkward spacing that disrupts the overall flow of the text, making it stand out as an alteration.
4. **Texture anomaly:** The texture of the tampered area may exhibit differences in pixelation and resolution when compared to the adjacent authentic text. This could manifest as a lack of clarity or distinctiveness that is noticeable upon closer inspection, suggesting it may not be a naturally occurring part of the document.
5. **Lighting inconsistency**: The lighting and shading on the tampered text "5" do not appear to match the surrounding authentic text. The illumination may look flat or inconsistent, contributing to the perception that this text has been altered rather than being part of the original document.

Figure 13: The prompt to query GPT-4o in the step 2.

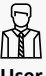

**User**

You are an expert good at analyzing tampered text images. You will be provided with the OCR information of the tampered area and two images, **the first is the tampered text image A and the second is the reference image B, with the tampered areas highlighted in the yellow box and the authentic areas darkened**. The tampered text **reads "5"**, and is located at the **top right** of the images A and B

You will also be given an analysis of the forged text, **which is usually unsatisfactory because of the uncertain tones and possible errors. Your task is to examine** each item of the provided analysis and find the inconclusive description (e.g. containing the words "could", "might" or "may"): For each inconclusive description, please check whether the artifacts described are obvious. If so, please rewrite the description to a confident one without the word "could", "might" or "may". If not, please delete the item. Please also delete the item if the content violates its title.

For example, given an input analysis as following:  Judgement basis
1. Edge artifacts: The edges around the tampered text "8X" may exhibit noticeable discontinuity when compared to the surrounding background. This might appear as a slightly jagged or irregular edge which doesn't seamlessly blend with the rest of the interface elements, indicating tampering.
2. Texture anomaly: The texture of the tampered text might appear slightly out of focus, overly sharp, or not as smoothly rendered as authentic text elements. Since these anomalies would contrast with the naturally rendered surrounding text, they can be indicative of manipulation.
3. Font inconsistency: The font used for "8X" could differ in subtle ways such as color, brightness, or style from the adjacent text. This inconsistency would suggest that the element does not match the visual identity of the rest of the display, pointing towards it being tampered.
4. Alignment inconsistency: Tampered text like "8X" might have spacing issues with neighboring text or be slightly misaligned with other elements on the same plane, deviating from the uniformity expected in a well-designed digital interface.
5. Text incoherence: The tampered word "8X" does not break the coherence of the sentence. However, if the surrounding context contained a different grammatical structure that rendered the word unnecessary or incorrectly placed, it could create confusion. In this case, "8X " fits grammatically, but the context of the surrounding text may be examined to determine overall coherence.
6. Lack of integration: The overall lack of integration of "8X" with the rest of the UI elements might be apparent. This could give it an appearance of being improperly overlaid or pasted onto the image, contrasting against the more naturally flowing and cohesive design of the authentic text.
**The expected response should be:**  Judgement basis
2. Texture anomaly: The texture of the tampered text appears slightly out of focus. Since this anomaly would contrast with the naturally rendered surrounding text, it can be indicative of manipulation.
3. Font inconsistency: The font used for "8X" differs in subtle ways such as color, brightness, or style from the adjacent text. This inconsistency would suggest that the element does not match the visual identity of the rest of the display, pointing towards it being tampered.
6. Lack of integration: The overall lack of integration of "8X" with the rest of the UI elements is apparent. This gives it an appearance of being improperly overlaid or pasted onto the image, contrasting against the more naturally flowing and cohesive design of the authentic text.

There are the words "could", "might" or "may" in the items titled "Edge Artifacts" and "Alignment Inconsistency", but there are no obvious edge artifacts or inconsistent alignment in the tampered text region. Therefore, these two items should be deleted. You should not delete the items just because they contain uncertain words such as "could" or "might". You should examine image A carefully and delete the uncertain descriptive items if the described anomalies actually do not exist. In addition, the content of the item entitled "Text incoherence" states that there is no text incoherence, which violates its title "Text incoherence" and should also be deleted.

**The input analysis is: (Providing the previous GPT-4o response)**

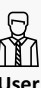

**User**

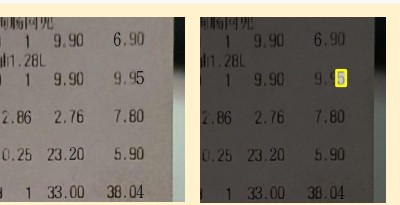

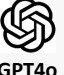

**GPT4o**

1. **Font inconsistency**: The font of the tampered text "5" is different in color and thickness when compared to the surrounding numbers. This variation draws attention to the tampered text, as it does not match the uniformity of the authentic text in size and style.
2. **Alignment inconsistency**: The placement of the tampered text "5" is slightly misaligned compared to the authentic text. This offset creates an awkward spacing that disrupts the overall flow of the text, making it stand out as an alteration.
3. **Texture anomaly:** The texture of the tampered area exhibits differences in pixelation and resolution when compared to the adjacent authentic text. This manifests as a lack of clarity or distinctiveness that is noticeable upon closer inspection, suggesting it is not a naturally occurring part of the document

Figure 14: The prompt to query GPT-4o in the step 3.

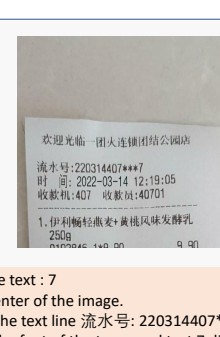 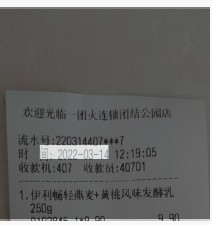

**Tampered Region:** the text : 7
**Absolute Position:** Center of the image.
**Qwen2.5VL-7B (Baseline), incorrect**
**Relative Position:** in the text line 流水号: 220314407**7
**Font Inconsistency**: The font of the tampered text 7 differs from the original text, particularly in terms of font size and character thickness. This suggests that the text 7 in this text line is tampered with.

**Tampered Region:** the text : 2022-03-14
**Absolute Position:** Bottom left of the image.
**Omni-IML (Ours), correct**
**Relative Position:** in the text line 时间: 2022-03-14 12:18:05
**Edge Artifacts**: The edges of the tampered text " 2022-03-14 " appear less defined compared to the surrounding text. There is slight blurriness or irregularities at the boundaries, indicating a mismatch with the background texture and color, which suggests tampering.
**Font Inconsistency**: The font of the tampered text is different in characteristics compared to the authentic text. This is observable in the color and thickness of the characters, which do not match the surrounding text, making the tampered text visually distinct.
**Alignment Inconsistency**: The "2022-03-14" has a vertical direction, which is inconsistent with the surrounding text, which disrupts the uniformity of the image style.

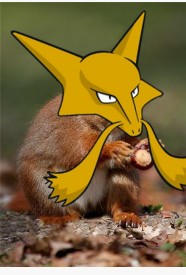 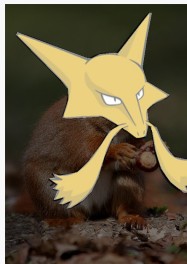

**Tampered Region:** The orange cartoon fox head with black edges.
**Absolute Position:** Center of the image.
**Qwen2.5VL-7B (Baseline), less comprehensive**
**Relative Position:** Over the body of the eating brown squirrel.
**Texture Inconsistency**: The cartoon fox's texture sharply contrasts with that of the brown squirrel and the background. While the fox features a smooth, cartoon-like texture, the squirrel exhibits a natural fur texture rich in depth and detail. This disparity creates a visually discordant effect.
**Lighting Inconsistency**: The lighting on the cartoon fox is inconsistent with the illumination present in the surrounding scene. The placement of shadows and highlights on the fox fails to align with the image's natural light source, resulting in an artificial look. Additionally, the fox lacks the realistic shadows that would naturally occur in the ground.
**Color Inconsistency**: The cartoon fox's colors are far more vibrant and uniform compared to the muted, varied tones of the squirrel and the background. This heightened saturation causes the fox to appear unnaturally distinct from the subdued color palette of the background.

**Tampered Region:** The orange cartoon fox head with pointed ears.
**Absolute Position:** Center of the image.
**Omni-IML (Ours), comprehensive**
**Relative Position:** Over the brown squirrel eating a nut.
**Edge Artifacts**: The edges of the cartoon fox head are too sharp and lack a smooth transition to the background. These artifacts indicate that the cartoon fox head has been tampered with.
**Texture Inconsistency**: The texture of the cartoon fox is markedly different from that of the brown squirrel and the background. The fox has a flat, cartoonish texture, while the squirrel has a natural fur texture that conveys depth and detail. This inconsistency creates a jarring visual effect.
**Lighting Inconsistency**: The lighting on the cartoon fox does not match the lighting conditions of the surrounding environment. The shadows and highlights on the fox do not align with the natural light source in the image, leading to an unnatural appearance. The fox lacks realistic shadowing that would typically occur in relation to the squirrel and the ground.
**Color Inconsistency**: The color saturation of the cartoon fox is more vibrant and uniform compared to the more subdued and varied colors of the squirrel and the background. This difference in saturation makes the fox stand out unnaturally against the more muted tones of the rest of the image.
**Depth Artifacts:** The depth of the cartoon fox is inconsistent with the squirrel. The fox is positioned on top of the squirrel, but the perspective and scale do not align properly, creating a sense of disorientation in spatial relationships.

Figure 15: Comparisons in image forgery interpretation.

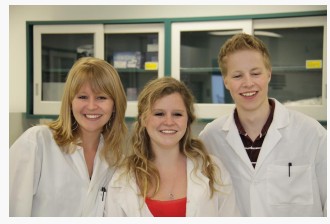 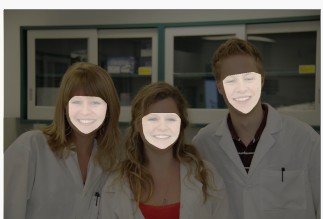

**Tampered Region 1**: The face of the woman with short brown hair wearing a white lab coat   **Qwen2.5VL-7B (Baseline), in**comprehensive
**Absolute Position:** Right side of the image.
**Relative Position:** Adjacent to the woman in the red shirt with long brown hair
**Edge Artifacts:** The transitions around the hairline and jaw display rough, unnatural edges that fail to seamlessly integrate with the surrounding areas.
**Texture Inconsistency:** The skin appears overly polished, missing the fine details like pores or blemishes, resulting in an artificial, plastic-like quality.
**Lighting Inconsistency:** The lighting on the man's face clashes with the illumination of the overall scene, creating an awkward, unnatural contrast.

**Tampered Region 2**: The face of the woman with long blonde hair dressed in a white lab coat
**Absolute Position:** Center of the image
**Relative Position:** Between the woman with short brown hair and another woman with short blonde hair
**Edge Artifacts:** The boundary between her hair and skin displays clear evidence of tampering, with visible gaps and abrupt transitions.
**Texture Inconsistency:** The skin texture alternates between being overly smooth and exhibiting strange, inconsistent patterns in certain spots.
**Lighting Inconsistency:** The lighting on the face significantly differs from that on the hair, suggesting that these two elements were edited independently before being merged.

**Tampered Region 1**: The face of the man with short brown hair and a white lab coat   **Omni-IML (Ours), comprehensive**
**Absolute Position:** Right side of the image.
**Relative Position:** To the right of the center woman with long brown hair and a red shirt
**Edge Artifacts:** The edges around the hairline and jawline are jagged and do not blend smoothly with the rest of the image.
**Texture Inconsistency:** The skin texture is overly smooth and lacks natural pores or blemishes, giving it an artificial appearance.
**Lighting Inconsistency:** The lighting on the face is inconsistent with the background, creating a stark contrast that highlights the tampering.
**High-level Artifacts:** The teeth show signs of irregular spacing and size, contributing to the unnatural look of the smile.

**Tampered Region 2**: The face of the left woman with blonde hair and a white lab coat
**Absolute Position:** Left side of the image
**Relative Position:** To the left of the center woman with curly hair wearing a red top
**Edge Artifacts:** The edges around the face, especially near the hairline, show clear signs of manipulation with jagged and unnatural lines.
**Texture Inconsistency:** The skin texture is overly smooth and lacks the natural pores, wrinkles, and imperfections typical of real human skin.
**Lighting Inconsistency:** The lighting on the face is inconsistent, with some areas appearing overly bright while others are unnaturally shadowed.
**High-level Artifacts:** The teeth show signs of blurring and lack the sharpness and detail expected in a real smile.

**Tampered Region 3:** The face of the center young woman with long blonde hair and a white lab coat
**Absolute Position:** Center of the image
**Relative Position:** Between the woman with short blonde hair and the man with short brown hair
**Edge Artifacts:** The edges around the face, particularly where the hair meets the skin, show clear signs of manipulation, with unnatural lines and discontinuities.
**Texture Inconsistency:** The skin texture is inconsistent, with some areas appearing overly smooth and others showing unnatural patterns.
**Lighting Inconsistency:** The lighting on the face does not align with the lighting on the hair, indicating that the face and hair were likely edited separately and then combined.
**Depth Artifacts:** The eyes and mouth appear overly simplified and lack the intricate details and depth that are characteristic of real human features.

Figure 16: Comparisons in image forgery interpretation.

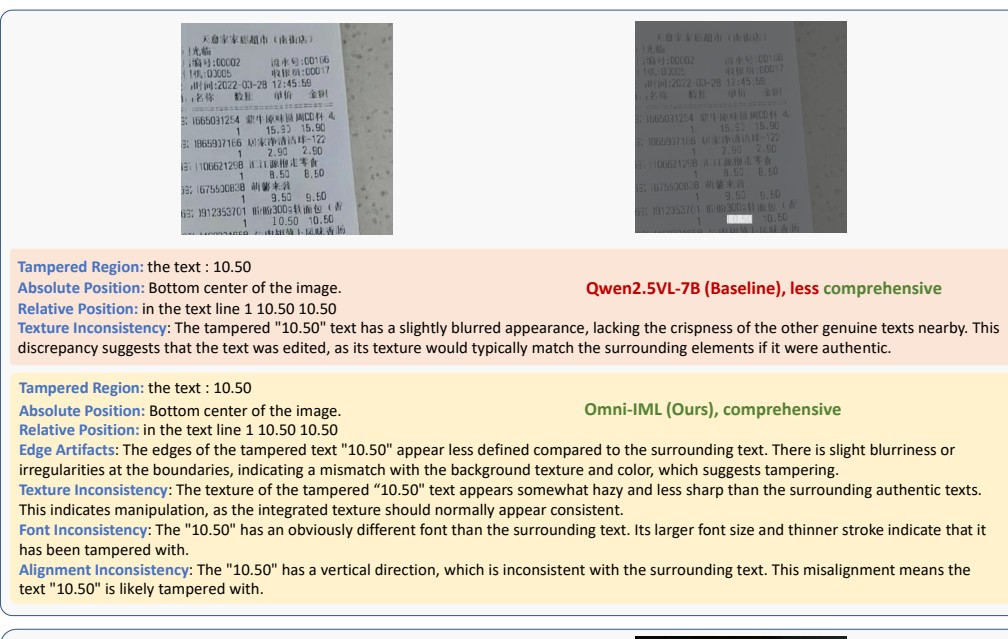

**Tampered Region:** the text : 10.50
**Absolute Position:** Bottom center of the image.
**Relative Position:** in the text line 1 10.50 10.50

**Qwen2.5VL-7B (Baseline), less comprehensive**

**Texture Inconsistency**: The tampered "10.50" text has a slightly blurred appearance, lacking the crispness of the other genuine texts nearby. This discrepancy suggests that the text was edited, as its texture would typically match the surrounding elements if it were authentic.

**Tampered Region:** the text : 10.50
**Absolute Position:** Bottom center of the image.
**Relative Position:** in the text line 1 10.50 10.50

**Omni-IML (Ours), comprehensive**

**Edge Artifacts**: The edges of the tampered text "10.50" appear less defined compared to the surrounding text. There is slight blurriness or irregularities at the boundaries, indicating a mismatch with the background texture and color, which suggests tampering.
**Texture Inconsistency**: The texture of the tampered "10.50" text appears somewhat hazy and less sharp than the surrounding authentic texts. This indicates manipulation, as the integrated texture should normally appear consistent.
**Font Inconsistency**: The "10.50" has an obviously different font than the surrounding text. Its larger font size and thinner stroke indicate that it has been tampered with.
**Alignment Inconsistency**: The "10.50" has a vertical direction, which is inconsistent with the surrounding text. This misalignment means the text "10.50" is likely tampered with.

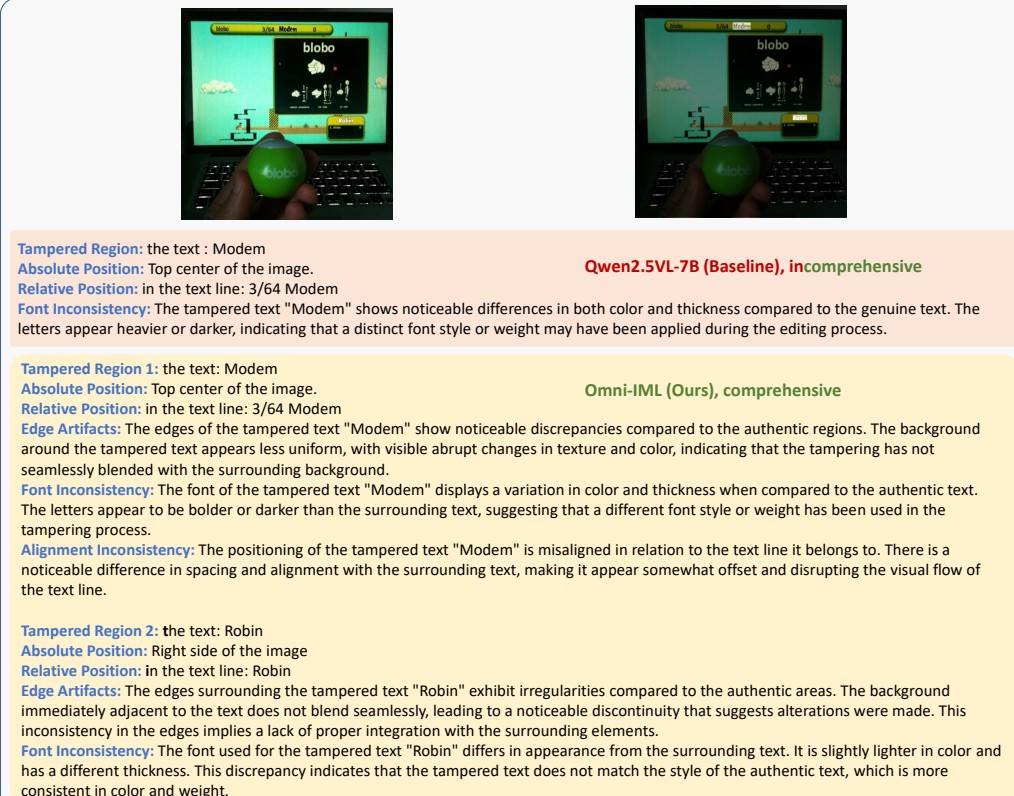

**Tampered Region:** the text : Modem
**Absolute Position:** Top center of the image.
**Relative Position:** in the text line: 3/64 Modem

**Qwen2.5VL-7B (Baseline), incomprehensive**

**Font Inconsistency:** The tampered text "Modem" shows noticeable differences in both color and thickness compared to the genuine text. The letters appear heavier or darker, indicating that a distinct font style or weight may have been applied during the editing process.

**Tampered Region 1:** the text: Modem
**Absolute Position:** Top center of the image.
**Relative Position:** in the text line: 3/64 Modem

**Omni-IML (Ours), comprehensive**

**Edge Artifacts:** The edges of the tampered text "Modem" show noticeable discrepancies compared to the authentic regions. The background around the tampered text appears less uniform, with visible abrupt changes in texture and color, indicating that the tampering has not seamlessly blended with the surrounding background.
**Font Inconsistency:** The font of the tampered text "Modem" displays a variation in color and thickness when compared to the authentic text. The letters appear to be bolder or darker than the surrounding text, suggesting that a different font style or weight has been used in the tampering process.
**Alignment Inconsistency:** The positioning of the tampered text "Modem" is misaligned in relation to the text line it belongs to. There is a noticeable difference in spacing and alignment with the surrounding text, making it appear somewhat offset and disrupting the visual flow of the text line.

**Tampered Region 2:** the text: Robin
**Absolute Position:** Right side of the image
**Relative Position:** in the text line: Robin

**Edge Artifacts:** The edges surrounding the tampered text "Robin" exhibit irregularities compared to the authentic areas. The background immediately adjacent to the text does not blend seamlessly, leading to a noticeable discontinuity that suggests alterations were made. This inconsistency in the edges implies a lack of proper integration with the surrounding elements.
**Font Inconsistency:** The font used for the tampered text "Robin" differs in appearance from the surrounding text. It is slightly lighter in color and has a different thickness. This discrepancy indicates that the tampered text does not match the style of the authentic text, which is more consistent in color and weight.

Figure 17: Comparisons in image forgery interpretation.

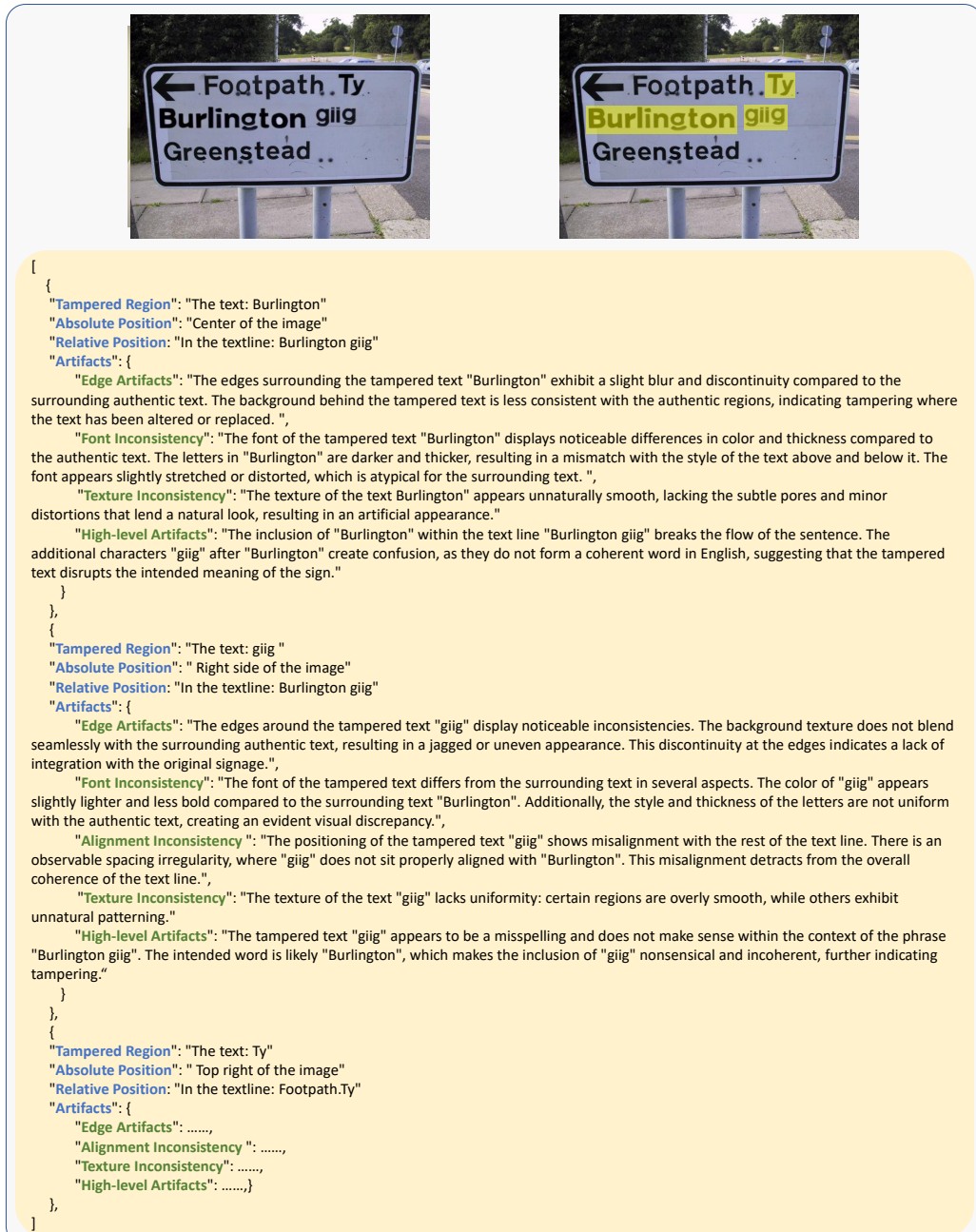

Figure 18: A representative sample to display our Omni-273k's structured annotation format.

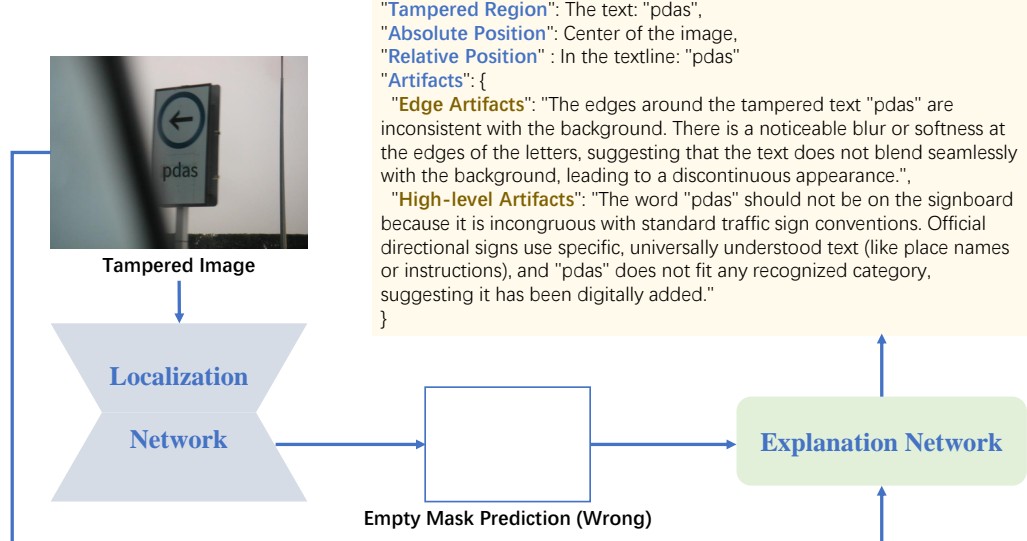

Figure 19: Our explanation module is designed to be robust against localization failures. It uses the localization mask as a guide but is not strictly bound by it. For example, if the localization module fails to detect a tampered region (i.e., predicts an empty mask), the explanation module can still independently identify the anomaly and report it, effectively correcting the initial error. In the predicted mask of this figure, white denotes real region, blue denotes tampered region.

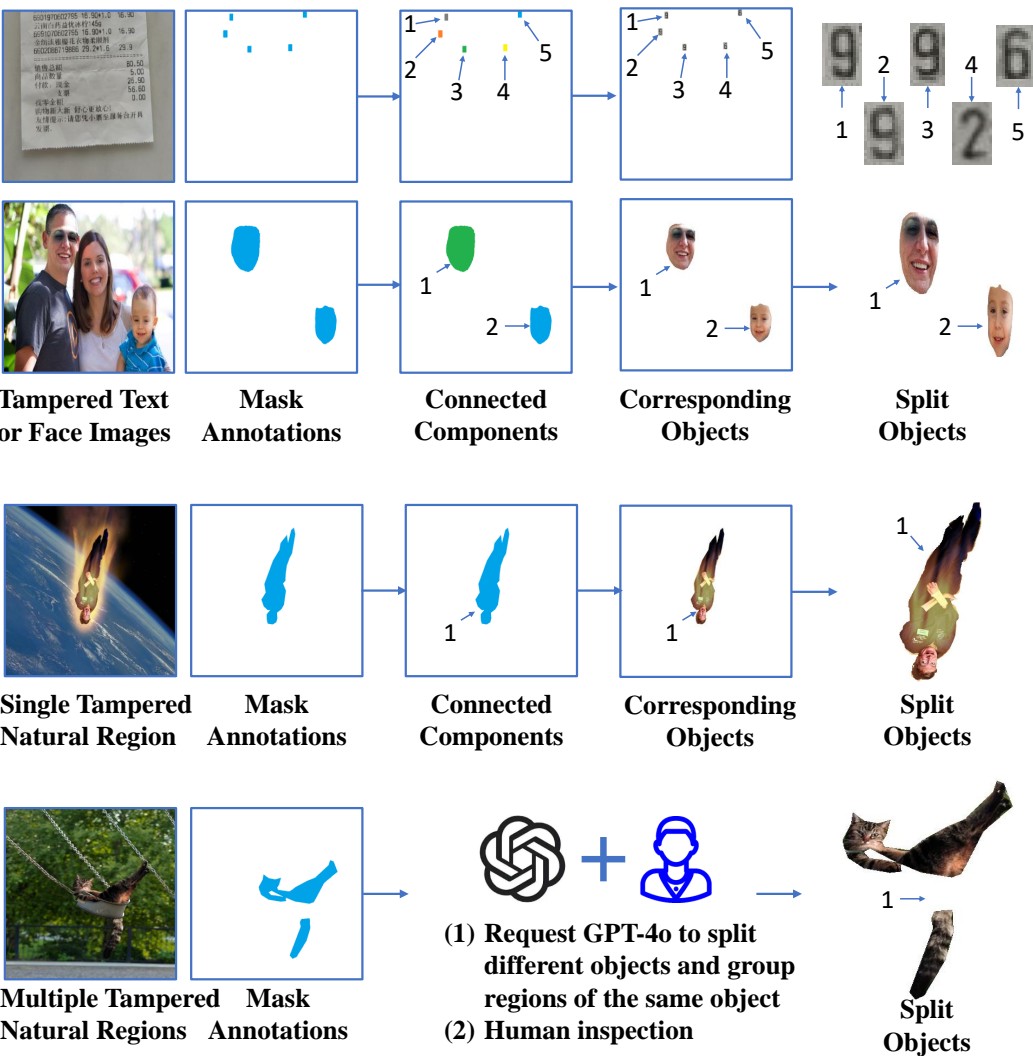

Figure 20: To split individual tampered objects, we employed a domain-specific strategy. For document, scene-text, and face images, where tampered objects typically consist of a single contiguous region, we used a connected-component analysis on the annotation masks. For natural images, where a single object may comprise multiple disconnected regions, we prompted GPT-4o to group mask fragments into semantic objects, followed by manual verification to ensure accuracy.

