# OpenReview forum: "Omni-IML: Towards Unified Interpretable Image Manipulation Localization"
_ICLR.cc/2026/Conference — ICLR 2026 Poster_

### Official Review · Reviewer_8wZG · 2025-10-22

**Soundness:** 2
**Presentation:** 3
**Contribution:** 2
**Rating:** 4
**Confidence:** 5

**Summary:**

This paper presents Omni-IML, a generalist model for Image Manipulation Localization. Its core methodological contributions are threefold: a Modal Gate Encoder for adaptive modality selection, a Dynamic Weight Decoder with task-aware filters, and an Anomaly Enhancement module that uses box supervision to learn task-agnostic features. The proposed single-model solution aims to unify multiple IML tasks, marking a promising step towards generalist models in image forensics.

**Strengths:**

Unified image forgery locazation is an study-worthy problem, and this paper also presents a promising solution for this problem. The paper is well organized and easy to follow.

**Weaknesses:**

1, My main concern is which is the primary focus of this paper?  a unified framework or interpretable? both hold many pages in methodology, but their connection is low. The explanation module integrates the IML results and finetune a MLLM for explanation text. However, the explanation seems does not benefit the IML accuracy in return, leading to the explanation section is relatively separate from the main focus. To me,  the localziation accuracy  is more important than explanation for IML problem, so it is reasonable and intuitive to promote localizaiton accuracy with explanation guidance.

2, Does the MLLM joinly tune with the localization networks?

3, how is the bbox supervision obtained in anmaly enhancement ?

4, During the dataset construction,  how to split the image to several tamperred objects from the binary mask? was it achieved with regulations or some algorithms?

5, As the localization mask has been given before generating the explanation, with this, the tampered region seem too simple. Besides, the artifacts description  is also too simple and general.

6, the parameter and inference time should be compared.

**Questions:**

see the weaknesses.

---

> ### Author Response · Authors · 2025-11-27
> **Response to Reviewer 8wZG (Part 1)**
>
> We sincerely thank Reviewer 8wZG for their valuable time.
>
> ## 1.a, Which is the primary focus of this paper? a unified framework or interpretable?
>
> * The unified framework serves as the foundation for this work (**primary**). The interpretability component is a crucial extension that enhances the model's comprehensive output, thus fulfilling both facets of our primary research goal.
> * Our paper's focus is to advance the omnipotence of image forensic models. We address this in two key dimensions:
> * On the **input side**: a unified framework capable of handling images from all major domains;
> * On the **output side**: complete forensic analysis, including: image-level classification, pixel-level localization of tampered regions, and a textual explanation of the observed artifacts for human comprehension and verification.
>
> ## 1.b The explanation seems does not benefit the IML accuracy in return, leading to the explanation section is relatively separate from the main focus?
>
> Our decoupled design for the localization and explanation modules is driven by two significant advantages:
> * (1) **Superior Localization Accuracy and Efficiency**: It is well-established that Multimodal Large Language Models (MLLMs) are not optimized for precise spatial tasks like predicting arbitrary-shaped segmentation masks [1, 2]. By using a dedicated, non-MLLM segmentation network for localization, we achieve significantly higher accuracy at a fraction of the inference cost. For instance, our model surpasses FakeShield [3] by a substantial 16.4 points in average IoU on the natural IML task (Table 2), while being nearly 9x faster (7.1 FPS vs. 0.4 FPS of FakeShield). This efficiency gain stems from our localization-only inference loop, whereas coupled approaches like FakeShield must invoke both non-MLLM and MLLM components for localization.
> * (2) **Increased Flexibility and Modularity**: Not all application scenarios require a textual explanation. As Reviewer 8wZG noted, "the localization accuracy is more important" for many users. Our decoupled architecture accommodates this by allowing the interpretation module (the MLLM) to be optionally excluded without affecting localization performance. This provides a faster, more lightweight deployment for scenarios where only localization is needed, while retaining the full explanation capability for those that require it.
>
> ## 1.c It is reasonable and intuitive to promote localization accuracy with explanation guidance.
>
> * Thank you for your suggestion. While our primary academic focus was on establishing the unified framework, we agree that exploring the synergy between explanation and localization is a valuable direction.
> * Following your recommendation, we used the model's textual output (object content and position) to guide a second-stage refinement. Specifically, we fed the explanations for natural/face images into an open-vocabulary segmentation model (DINO-X [4]) and the explanations for document/scene-text images into an OCR tool (OCRSpace) to generate refined localization masks.
> * The results, now included as Table 18 (Section H.2), show a consistent improvement in localization accuracy. This confirms that explanation can indeed be used to refine localization and opens a promising avenue for future work.
>
> |Method|Natural|Document|SceneText|Face|Avg.|
> |-|-|-|-|-|-|
> |Ours (original)|.612|.766|.610|.923|.728|
> |+ Explanation Refinement|.636|.774|.641|.925|.744|
>
> ## 2, Does the MLLM jointly tune with the localization networks?
>
> * The MLLM and localization networks are trained entirely independently.
> * This is clarified at the start of Section 5 in our original paper (Lines 354 to 355 of our original paper, or Lines 356 to 357 of our revised paper). This ensures a fair comparison with existing localization models (Tables 2, 3, 4).
>
> ## 3, How is the bbox supervision obtained in anomaly enhancement?
>
> * This bbox supervision is obtained from the ground-truth masks by computing the bounding box that encloses each connected component within the forgery mask.
> * This process was clarified in Appendix Section C.2 (Line 911 of our original paper), and was illustrated at the center-top of Figure 2 (titled “Bounding Box Supervision”).

---

> ### Author Response · Authors · 2025-11-27
> **Response to Reviewer 8wZG (Part 2)**
>
> ## 4, During the dataset construction, how to split the image to several tampered objects from the binary mask? was it achieved with regulations or some algorithms?
>
> * For most images, we treat each connected component in the binary mask as a separate tampered object.
> * This was clarified in Appendix Section D.1 of our original paper (Lines 963 to 966).
> * For the NIST16, IMD20, and MIML datasets, multiple disconnected regions can belong to a single tampered object. Therefore, we leverage GPT-4o's reasoning capabilities by providing the image with all tampered regions highlighted and prompting it to describe each distinct object. This process effectively groups the related regions.
> * This was clarified in Lines 997 to 1002 of our original paper, and we have now added Figure 20 to the Appendix to illustrate this procedure.
>
> ## 5, As the localization mask has been given before generating the explanation, with this, the tampered region seem too simple. Besides, the artifacts description is also too simple and general.
>
> * **On the Localization Mask:** This binary mask format is the standard and established method in the image manipulation localization field [5, 6, 7]. Our model's ability to predict these masks for arbitrary shapes and quantities (as shown in Figures 10-12 and 15-17) is a key strength, it is not "too simple" or a weakness of our paper.
> * **On the Artifact Descriptions:** The artifacts descriptions in Figures 1 and 2 are high-level summaries to indicate our model's interpretation ability.
>
> * Full artifact descriptions can be found in Figures 15 to 17 of our original paper. Our artifacts description is designed to be concise to balance the comprehensiveness and inference time.
> * It can clearly and comprehensively explain the content, position and key artifact bases with its concise paragraphs.
> * We avoid too long paragraphs, which can significantly hurt readers' patience and increase model inference time in real-world applications.
> * It is also notable that almost all related works [3, 8] in image artifact description adopt the same level of description conciseness and paragraph length.
>
>
> ## 6, the parameter and inference time should be compared.
>
> * For the localization task, we already included a comparison of model parameters and inference speed in Table 16.
> * As shown, our localization module's complexity is comparable to that of other state-of-the-art specialized methods.
> * **Crucially**, in a real-world scenario requiring coverage of all four domains, a user would need four separate specialist models plus a domain classifier. Our single, unified model handles all domains, making it significantly more efficient in terms of both total parameters and deployment complexity, while achieving superior performance.
> * For explanation prediction, since we do not change the MLLM structure in interpretation module, the parameter and inference time are **identical** to the base MLLM (e.g. Qwen2.5-VL 3B).
>
> [1] Lan M, Chen C, Zhou Y, et al. Text4Seg: Reimagining Image Segmentation as Text Generation[C]//The Thirteenth International Conference on Learning Representations.
>
> [2] Su Y, Zhang H, Li S, et al. Patch-as-Decodable-Token: Towards Unified Multi-Modal Vision Tasks in MLLMs[J]. arXiv preprint arXiv:2510.01954, 2025.
>
> [3] Xu Z, Zhang X, Li R, et al. Fakeshield: Explainable image forgery detection and localization via multi-modal large language models[J]. arXiv preprint arXiv:2410.02761, 2024.
>
> [4] Ren T, Chen Y, Jiang Q, et al. Dino-x: A unified vision model for open-world object detection and understanding[J]. arXiv preprint arXiv:2411.14347, 2024.
>
> [5] Guillaro F, Cozzolino D, Sud A, et al. Trufor: Leveraging all-round clues for trustworthy image forgery detection and localization[C]//Proceedings of the IEEE/CVF conference on computer vision and pattern recognition. 2023: 20606-20615.
>
> [6] Kong C, Luo A, Wang S, et al. Pixel-inconsistency modeling for image manipulation localization[J]. IEEE Transactions on Pattern Analysis and Machine Intelligence, 2025.
>
> [7] Wong K, Zhou J, Wu H, et al. ADCD-Net: Robust Document Image Forgery Localization via Adaptive DCT Feature and Hierarchical Content Disentanglement[C]//Proceedings of the IEEE/CVF International Conference on Computer Vision. 2025: 19280-19289.
>
> [8] Huang Z, Hu J, Li X, et al. Sida: Social media image deepfake detection, localization and explanation with large multimodal model[C]//Proceedings of the Computer Vision and Pattern Recognition Conference. 2025: 28831-28841.

---

### Official Review · Reviewer_FDt3 · 2025-10-27

**Soundness:** 3
**Presentation:** 3
**Contribution:** 2
**Rating:** 6
**Confidence:** 3

**Summary:**

The paper introduces Omni-IML, a unified model for interpretable image manipulation localization (IML) across four major domains: natural images, documents, faces, and scene texts. It addresses the limitations of task-specific IML models by proposing a generalist architecture with key components: a Modal Gate Encoder for adaptive modality selection (vision-only or vision+frequency), a Dynamic Weight Decoder for sample-adaptive filtering, and an Anomaly Enhancement module using bounding box supervision to highlight tampered regions. Additionally, the authors construct Omni-273k, a large dataset with 273k samples annotated via a chain-of-thoughts (CoT) pipeline using GPT-4o for natural language descriptions of tampering artifacts. An interpretation module based on a multimodal large language model (MLLM) is introduced to generate textual explanations.

**Strengths:**

1. Unified Generalist Approach: This is the first work to effectively unify IML across diverse domains (natural, document, face, scene text) in a single model, achieving SOTA results on multiple benchmarks. It provides a practical solution for real-world deployment, reducing the need for multiple specialized models and addressing high maintenance costs.

2. Large-Scale Interpretable Dataset: Omni-273k is a comprehensive, high-quality dataset covering all four IML domains, with structured JSON annotations for tampered content, positions, and artifacts. The CoT annotation pipeline improves quality over prior single-prompt methods, enabling finer-grained evaluation and supporting interpretable forensics.

3. Effective Architectural Innovations for Generalization: Components like the Modal Gate (for modality adaptation) and Dynamic Weight Decoder (for handling diverse tampering clues) enable robust performance across tasks. The Anomaly Enhancement module leverages box supervision to enhance task-agnostic features, and the simple interpretation module (using visual prompts) boosts reliability without altering the MLLM structure.

4. Strong Empirical Validation: Extensive experiments on representative benchmarks show minimal performance degradation from joint training, with Omni-IML outperforming domain-specific SOTAs. The work also highlights interpretability, a underexplored aspect in unified IML.

**Weaknesses:**

1. The CoT annotation pipeline, while effective, feels incremental and assembled from existing ideas (e.g., step-by-step prompting and self-examination in GPT-4o). It may not be necessary for low-level tampering artifacts (e.g., texture or edge inconsistencies), which are often straightforward and do not require repeated reasoning or context-heavy analysis, potentially overcomplicating the process without strong justification.

2.  Modules like the Modal Gate and Dynamic Weight Decoder rely on well-established techniques (e.g., gating mechanisms for modality selection, dynamic filtering akin to those in dynamic networks). This makes the contributions appear more as combinations of prior art rather than groundbreaking innovations, reducing the perceived technical depth.

3. The focus on low-level artifacts (e.g., font/alignment inconsistencies in documents) might limit generalizability to emerging tampering methods, such as those from advanced generative models. The evaluation could benefit from more diverse, real-world scenarios beyond the curated datasets.

4. While the CoT pipeline is claimed to improve annotation quality, there is insufficient evidence (e.g., human evaluations or comparisons to non-CoT baselines) to demonstrate its necessity, especially for simpler domains where context information is weak.

5. SThe interpretation module operates as a standalone component, not integrated with the localization process (e.g., the Dynamic Weight Decoder). This separation may reduce the model's ability to provide cohesive, context-aware explanations, potentially missing opportunities for joint optimization of localization and interpretability.

**Questions:**

1.  Why is repeated reasoning (e.g., self-examination) necessary for low-level artifacts like texture inconsistencies, which seem more perceptual than contextual? Could a simpler single-prompt approach suffice for document or scene text domains, and what ablation results support the added complexity?
2.  How do the Modal Gate and Dynamic Weight Decoder differ substantially from existing gated or dynamic mechanisms (e.g., in multimodal fusion or dynamic convolutions)? Can you provide comparisons to show that these are not just incremental adaptations?

3. In cases where tampering is subtle and low-level (e.g., no strong semantic context), does the visual prompt risk introducing hallucinations in the MLLM? What metrics were used to evaluate this, and how does it perform on edge cases like multi-instance documents?

4. Given the incremental nature of the pipeline, how does Omni-IML handle novel tampering methods not seen in the training datasets, such as those from future diffusion models? Are there cross-domain transfer experiments to validate true generalization?

5. Since the interpretation module is separate from the localization pipeline, how does this design choice impact the model's ability to provide consistent and context-aware artifact descriptions? Would integrating interpretability into the Dynamic Weight Decoder or other core components improve overall performance or coherence?

---

> ### Author Response · Authors · 2025-11-27
> **Response to Reviewer FDt3 (Part 1)**
>
> We sincerely thank Reviewer FDt3 for their valuable time, thorough review and constructive feedback.
>
> ## Response to Weakness 1, 4 & Question 1 (On Our CoT Pipeline's Necessity and Comparison)
> **1.a On the Necessity of Repeated Reasoning:**
>
> Our iterative reasoning process is crucial for two reasons, as analyzed in lines 244-249:
>
> * **Preventing Attentional Distraction**: When multiple forgeries are present, a single-pass analysis can cause the model to misattribute artifacts. Our iterative approach ensures focused, accurate, and localized analysis for each potential manipulation.
> * **Counteracting MLLM Compliance Bias**: MLLMs often exhibit compliance bias. Open-ended prompts can lead to under-reporting, while checklist-style prompts can induce "hallucinations." Our self-examination step acts as a refinement filter, compelling the MLLM to scrutinize its initial findings, discard spurious detections, and converge on high-confidence, factually-grounded evidence.
>
> **1.b Comparison with non-CoT method:**
> * To quantify the necessity of our CoT approach, we randomly selected 400 test samples from each domain and generated tampered-region explanations using the previous non-CoT approach (as in FakeShield [1]). We then compared these non-CoT outputs with our recorded CoT responses, evaluating anomaly descriptions both against our manually cleaned ground-truth using the same metrics as Table 6.
> * The results, shown in the table below, clearly demonstrate the advantages of our CoT reasoning — particularly in multi-object scenarios, such as text and face images. The results have been updated to our revised paper (Table 20. Appendix Section H.4). "A.D." denotes "Artifacts Description"
>
> |Method|Doc.-A.D.-Acc|Doc.-A.D.-MRB|S.T.-A.D.-Acc|S.T.-A.D.-MRB|Nat.-A.D.-Acc|Nat.-A.D.-MRB|Face-A.D.-Acc|Face-A.D.-MRB|
> |-|-|-|-|-|-|-|-|-|
> |non-CoT|.673|.815|.521|.797|.802|.870|.695|.924|
> |CoT (Ours)|.949|.892|.968|.903|.966|.921|.989|.970|
>
> ## Response to Weakness 2 & Question 2 (On the Novelty of Our Modules)
>
> Our modules introduce key architectural innovations over prior work. As explicitly acknowledged by Reviewers 91UG and d5Rc, and validated by the new ablations below, our designs are both novel and effective.
>
> **2.a On the novelty of the Modal Gate:**
>
> Our Modal Gate is fundamentally different from existing fusion mechanisms in two key aspects:
> * **Binary Hard Gate**: Unlike prior work using soft attention for feature fusion, our module employs a binary hard gate. This decisive mechanism proves significantly more effective, as validated by our ablation study in Table 8 of original our paper.
> * **Explicit Supervision of Fusion Weights**: Unlike prior work that learns fusion weights implicitly via the final model loss, we introduce a direct supervision mechanism. Our method generates a supervisory signal for the fusion weights by identifying which modality produces a more accurate coarse prediction. Ablation studies in Table 5 of original our paper confirm the effectiveness of this explicit supervision strategy (detailed in Appendix C.1, Lines 855-863).
> * **Prediction-Informed Gating**: Conventional gates fuse features directly. In contrast, our Modal Gate makes its decision by also considering the confidence of coarse predictions derived from each modality. This allows the gate to assess the reliability of each feature set for the specific input, leading to a more intelligent and effective fusion. We have added a new ablation study (see table below) that confirms this design choice yields superior performance (IoU). The results have been updated to our revised paper (Table 21. Appendix Section H.5)
>
> |Ablation|Natural|Document|SceneText|Face|Avg.|
> |-|-|-|-|-|-|
> |without coarse prediction features|.560|.742|.602|.915|.705|
> |with coarse prediction features|.612|.766|.610|.923|.728|
>
> **2.b On the novelty of the Dynamic Weight Decoder:**
>
> * Our design overcomes a key limitation of standard dynamic convolutions. While conventional methods generate weights using only local features from their receptive field, our Dynamic Weight Decoder incorporates **an additional global context vector (Vg in Figure 6).**
> * This vector, which summarizes the entire feature map, informs the decoder about the global image type (e.g., natural vs. document).
> * This global awareness enables the generation of more specialized and effective dynamic weights, a clear departure from prior methods.
> * The results have been updated to our revised paper (Table 22. Appendix Section H.6)
>
> |Ablation|Natural|Document|SceneText|Face|Avg.|
> |-|-|-|-|-|-|
> |without global vector Vg|.565|.721|.584|.918|.697|
> |with global vector Vg|.612|.766|.610|.923|.728|

---

> ### Author Response · Authors · 2025-11-27
> **Response to Reviewer FDt3 (Part 2)**
>
> ## Response to Question 3 (Risk of Hallucinations from the Visual Prompt)
>
> * While incorrect localization predictions can occasionally introduce hallucinations in the MLLM, the benefits of our visual prompts far outweigh this weakness. Our prompts reduce hallucinations in many more cases than they cause them.
> * Evidence: In Table 6, Qwen2.5-VL 7B (Qw.7B) with our visual prompt shows substantial gains over the same model without prompts — e.g., Tampered Text recognition (OCR) improves from 0.312 to 0.653 — confirming the prompt’s effectiveness in reducing hallucinations, especially for multi-instance documents.
>
> * Notably, our explanation module is not strictly bound by the localization output. It uses the mask as a guide, but the MLLM can independently identify and override failures from the localization stage.
> * We provide a qualitative example of this error-correction capability in Figure 19 of our revised paper.
>
>
> ## Response to Weakness 3 & Question 4 (On Generalization to Novel Tampering Methods)
>
> Our model demonstrates strong generalization to unseen tampering methods, a capability already validated by our existing experiments:
>
> * **Unseen tampering methods on non-text images**: Our training set contains no diffusion-generated forgeries. Yet, as shown in Table 2, our model significantly outperforms all baselines on the CocoGlide dataset, which consists entirely of images manipulated by modern diffusion models.
> * **Unseen tampering methods on text images**: In Table 3, the OSTF benchmark contains seven tampering methods absent from our training set. Our model surpasses the next-best method by over 10 IoU points.
>
> These results provide direct evidence of the model's ability to generalize beyond its training distribution to novel, real-world threats.
>
> ## Response to Weakness 5 & Question 5 (Module Independence)
>
> **5.a On the separation of localization and explanation modules:**
>
> We intentionally separate localization and explanation to maximize accuracy, efficiency, and deployment flexibility:
> * **Accuracy & Efficiency:** Using a dedicated segmentation network avoids MLLM overhead, delivering state-of-the-art IoU and ~9× faster localization compared to FakeShield (7.1 FPS vs. 0.4 FPS) [1].
> * **Flexibility:** The localization module can run standalone for speed-critical tasks, or be paired with the MLLM where explanations are needed, with no trade-off in localization accuracy.
>
> **5.b Possibility of integrating interpretability into core components:**
>
> * Embedding interpretability directly into localization modules would require MLLMs, dramatically increasing model size and latency for the localization prediction, hurting its flexibility.
> * Instead, we improve performance and coherence through explanation-driven localization refinement: we implemented a new experiment to demonstrate an explanation-to-localization feedback loop.
> * We fed the textual explanations into an open-vocabulary segmentation model (DINO-X [2]) or an OCR tool (OCRSpace from GitHub) to refine the initial masks.
> * As shown below and in Table 18 (Section H.2), this refinement yields a consistent IoU improvement, confirming the potential of this promising research direction.
>
> |Method|Natural|Document|SceneText|Face|Avg.|
> |-|-|-|-|-|-|
> |Ours (original)|.612|.766|.610|.923|.728|
> |+ Explanation Refinement|.636|.774|.641|.925|.744|
>
> [1] Xu Z, Zhang X, Li R, et al. Fakeshield: Explainable image forgery detection and localization via multi-modal large language models[J]. arXiv preprint arXiv:2410.02761, 2024.
>
> [2] Ren T, Chen Y, Jiang Q, et al. Dino-x: A unified vision model for open-world object detection and understanding[J]. arXiv preprint arXiv:2411.14347, 2024.

---

### Official Review · Reviewer_d5Rc · 2025-10-27

**Soundness:** 3
**Presentation:** 2
**Contribution:** 3
**Rating:** 6
**Confidence:** 3

**Summary:**

The paper proposes Omni-IML, aiming to unify four major image manipulation localization (IML) tasks—natural images, documents, faces, and scene text—within a single general model. The core architecture comprises three components: a Modal Gate Encoder (sample-adaptive modality selection), a Dynamic Weight Decoder (sample-adaptive decoder filters), and Anomaly Enhancement (box-level supervision to boost contrast of manipulated regions during training). In addition, the paper presents an explanation module that overlays the predicted mask as a visual prompt to an MLLM, and constructs a large-scale dataset with structured text annotations, Omni-273k. Experiments claim single-model SOTA across multiple benchmarks in the four IML categories with minimal degradation under joint training. The authors commit to releasing code and data.

**Strengths:**

1. Introduces the first practical general IML model, Omni-IML, addressing the long-standing issue that existing methods suffer significant performance drops under joint training.
2. Novel and effective architectural design: the Modal Gate Encoder elegantly resolves the “double-edged sword” of frequency features—very helpful for document images but potentially noisy for natural images—by adaptively selecting the optimal modality. The Dynamic Weight Decoder adjusts filters dynamically to cope with diverse manipulation types. The Anomaly Enhancement module leverages box supervision during training to strengthen anomaly cues without adding inference cost, which is a neat design.
3. Builds a large-scale, high-quality, and comprehensive dataset, Omni-273k, providing stronger foundations for the field, along with a novel automatic annotation pipeline.
4. Extensive experiments validate the effectiveness of the proposed method.

**Weaknesses:**

1. The main contribution is performance, but it comes with higher computational cost. As shown in Table 16, Omni-IML (152M parameters) is larger than most methods (e.g., DTD 66M, SparseViT 50M) and its inference speed (7.1 FPS) is slower than most methods. Considering practical deployment, such high computational cost could be a barrier.
2. The localization and explanation modules are independent. The explanation module simply uses the localization mask as a visual prompt; there is no end-to-end or error-correction feedback mechanism. If the localization module misses regions, the explanation module receives no prompt there, likely leading to explanation failure. This loose coupling limits the system from being a truly end-to-end interpretable solution.
3. The paper states a cross-domain sampling ratio of 10:5:2:20 for natural, document, face, and scene-text images, respectively. However, the scene-text training set has only 229 images yet receives the highest weight. Such extreme re-sampling may introduce generalization/bias side effects. The paper currently lacks sensitivity analyses or rationale justifying this choice.
4. Numerous writing and typesetting issues hinder readability. For example, typos in the first line of the intro (“Mmanipulated”, “servious”), the title of Figure 5 (“Anomanly Enhancement”), and “Singe and multiple” in Table 1.

**Questions:**

1. Can the overall architecture be light-weighted for deployment (e.g., smaller backbones, distillation, pruning)? How does the end-to-end latency compare to methods with similar or larger model sizes (e.g., PIM, UPOCR) under the same input resolution, given that the paper reports FPS only?
2. Can the localization and explanation modules be jointly trained, and what are the effects? How does the explanation module behave under missed detections? Any attempts at feedback from explanation to localization for error correction?
3. Can you provide justification for the 10:5:2:20 cross-domain sampling ratio? Given the very small number of scene-text samples, does heavy re-sampling cause overfitting or bias in practice?

**Details Of Ethics Concerns:**

N.A.

---

> ### Author Response · Authors · 2025-11-27
> **Response to Reviewer d5Rc (Part 1)**
>
> We sincerely thank Reviewer d5Rc for their valuable time, thorough review and constructive feedback.
>
> ## Response to Weakness 1 & Question 1 (Computational Cost & Latency)
> **1.a On Computational Cost:**
>
> * We respectfully argue that our model's cost is highly efficient when evaluated as a unified framework. While our 152M parameters seem large compared to specialized models, our model is substantially more lightweight than other unified approaches like UPOCR (192M) [1] and All-in-One (457M) [2].
>
> * Critically, our single model handles all major tampering types end-to-end. A practical system built from specialized models would require deploying at least four separate networks plus a classifier, incurring a far greater cost in both parameter storage and cumulative inference time. Therefore, for real-world deployment, our unified approach is significantly more efficient.
>
> **1.b On Latency Comparison:**
>
>  For a fair comparison in Table 16, all models were benchmarked on the same machine using a uniform 512x512 input resolution.
>
> **1.c On a Lightweight Model:**
>
> We thank the reviewer for this valuable suggestion and have implemented a lightweight version of our model. As shown below and in Table 17 (Appendix Section H.1), replacing ConvNeXt-Base with ConvNeXt-Small results in a 25% size reduction and a 31% speed increase, with only a negligible drop in average performance. This demonstrates a compelling efficiency-performance trade-off.
>
> |Backbone|Size|FPS|Natural|Document|SceneText|Face|Avg.|
> |-|-|-|-|-|-|-|-|
> |ConvNeXt-Base|152M|7.1|.612|.766|.610|.923|.728|
> |ConvNeXt-Small|113M|9.3|.590|.745|.587|.919|.710|
>
> ## Response to Weakness 2 & Question 2 (Module Independence)
> **2.a On Independent Modules:**
>
> The separation of localization and explanation is a deliberate design choice that yields superior accuracy, efficiency, and flexibility.
>
> * **Accuracy & Efficiency**: Using a dedicated segmentation network allows our model to achieve state-of-the-art accuracy while being vastly more efficient. For instance, our model outperforms FakeShield's localization by 16.4 IoU and is nearly 9x faster (7.1 vs. 0.4 FPS) [3].
> * **Flexibility**: Users can deploy the lightweight localization network for speed-critical tasks or include the MLLM for applications requiring explanations, without performance trade-offs.
> * While incorrect localization can lead to explanation failure, its positive impact on explanation far outweighs potential drawbacks. **Evidence:** In Table 6, Qwen2.5-VL 7B (Qw.7B) with our visual prompt shows substantial gains over the same model without prompts — e.g., Tampered Text recognition (OCR) improves from 0.312 to 0.653 — confirming the prompt’s effectiveness in reducing hallucinations, especially for multi-instance documents.
> * Importantly, our system can achieve **error-correction** by leveraging explanations to improve localization, as detailed in our response 2.d.
>
> **2.b On Joint Training:**
>
> Thank you for your suggestion. Joint end-to-end training is infeasible with the current architecture due to the non-differentiable binarization of the localization mask. Exploring differentiable approximations for this connection is a promising direction for future work.
>
> **2.c Behavior on Missed Detections:**
>
> This highlights a key strength: our explanation module is not strictly bound by the localization output. It uses the mask as a guide, but the MLLM can independently identify and override failures from the localization stage. We provide a qualitative example of this error-correction capability in Figure 19 of our revised paper.
>
> **2.d Feedback from Explanation to Localization:**
>
> Inspired by this valuable suggestion, we implemented a new experiment to demonstrate an explanation-to-localization feedback loop. We fed the textual explanations into an open-vocabulary segmentation model (DINO-X [3]) or an OCR tool (OCRSpace from GitHub) to refine the initial masks. As shown below and in Table 18 (Appendix Section H.2), this refinement yields a consistent IoU improvement, confirming the potential of this promising research direction.
>
> |Method|Natural|Document|SceneText|Face|Avg.|
> |-|-|-|-|-|-|
> |Ours (original)|.612|.766|.610|.923|.728|
> |+ Explanation Refinement|.636|.774|.641|.925|.744|

---

> ### Author Response · Authors · 2025-11-27
> **Response to Reviewer d5Rc (Part 2)**
>
> ## Response to Weakness 3 & Question 3 (Sampling Ratio)
>
> * To address concerns regarding overfitting, we clarify that the sampling probability for each domain in each batch is proportional to (dataset size × sampling ratio).
> * Thus, even with a high sampling ratio for the small scene-text dataset, its actual selection probability remains balanced, effectively mitigating overfitting.
> * Our sampling ratio is empirically derived from two principles: task difficulty and image resolution, as detailed in the accompanying table. We account for resolution, as larger images (like those in the scene-text domain) yield more potential training crops.
>
> | |Natural|Document|SceneText|Face|
> |-|-|-|-|-|
> |Task Difficulty Factor (TDF)|5|5|5|1|
> |Average Image Resolution (AIR)|724|517|1090|873|
> |Sampling Ratio=Round(TDF*(AIR/512)^2)|10|5|20|2|
>
> * This principled approach is validated by a new ablation study (presented in Appendix Section H.3, Table 19), which confirms our chosen ratio is near-optimal and robustly balances against both overfitting and underfitting.
>
> |Sampling Ratio (Nat:Doc:Face:SceneText)|Natural|Document|Face|SceneText|Avg.|
> |-|-|-|-|-|-|
> |10:5:2:20 (Original)|.612|.766|.923|.610|.728|
> |10:5:10:20 (More face)|.603|.754|.925|.596|.719|
> |10:5:2:50 (More scene text)|.611|.766|.923|.601|.725|
> |10:5:2:5 (Less scene text)|.614|.764|.923|.549|.712|
>
>
> ## 4. Writing typos
> * We sincerely thank the reviewer for highlighting these issues.
>
> * The manuscript has undergone meticulous proofreading by **multiple human reviewers** and was thoroughly checked, word-by-word, using **professional typo-checking tools (e.g., Aspell)** on multiple occasions. **We are confident that all typos and typesetting errors have been fully addressed** and corrected in our revised version, **which has already been uploaded**.
>
> * To provide an even higher level of assurance, we further utilized multiple large language models (e.g., Gemini 3 and GPT-5) to conduct an additional comprehensive check. **All these models consistently confirmed the absence of any typographical or typesetting errors in the updated manuscript**.
>
> * **The revised manuscript, which has been thoroughly proofread, has now been uploaded.**

---

### Official Review · Reviewer_91UG · 2025-10-30

**Soundness:** 4
**Presentation:** 3
**Contribution:** 4
**Rating:** 8
**Confidence:** 4

**Summary:**

This paper introduces an unified model named Omni-IML for the localization and interpretation of image manipulation. Three novel modules are proposed in the Omni-IML, effectively addressing the task conflicts in the unified training. An interpretation module is proposed to improve model’s forgery interpretation accuracy by prompting the multi-modal large language model with the localization model’s prediction. The proposed Omni-IML achieves state-of-the-art average performance across four major image manipulation localization tasks.
This paper also contributes the Omni-273k dataset to support better training on image forgery interpretation, it is initially constructed by a chain-of-thought pipeline, effectively reducing human-washing cost. The proposed structured label format enables detailed and reasonable evaluation.

**Strengths:**

Unified generalist model across four IML domains
- A single model achieves strong generalization and SOTA performance on natural, document, face, and scene text tasks without per-task tuning.
- Addresses fragmentation and lack of generalization in prior task-specific solutions with a well-motivated unified design.

Sample-adaptive architecture design
- Modal Gate Encoder adaptively selects the optimal modality (RGB vs frequency+vision) per sample.
- Dynamic Weight Decoder introduces per-sample dynamic filters to mitigate joint-training degradation and improve feature fusion of low- and high-level features.
- Anomaly Enhancement module is cleverly designed to enhance forged-region contrast using box supervision, with empirical support in ablations.
- Visual prompts assist correct tampered-region identification, yielding more accurate artifact descriptions compared to prior methods.

The Omni-273k dataset as a substantive contribution
- Large-scale coverage across four major IML tasks, enabling comprehensive training and evaluation.
- Structured JSON annotations enable fine-grained, reasonable evaluation and analysis, improving over unstructured labels in prior work.
- Chain-of-thought automatic annotation pipeline enhances data availability and quality.

Extensive experimental validation
- Thorough experiments, ablation studies, and comparisons convincingly demonstrate the effectiveness of each proposed module and the unified system.
- Results consistently verify performance and generalization across diverse tasks.

Technical soundness and clarity
- Methodology is technically sound, with coherent component design and integration.
- Writing and paper structure are clear and easy to understand.

**Weaknesses:**

- Regarding the Modal Gate, other auxiliary modalities, except for frequency may produce better results. Also, a soft gate weighting fusion may be better than the hard gate in the current version.
- The model’s parameter size and computational burden are relatively heavy; it includes a multi-modal large model in both training and inference.
- The font size figure2 and figure3 is too small. It is suggested to simplify the content and enlarge the font size.

**Questions:**

See the concerns raised in the 'Weaknesses' section.

---

> ### Author Response · Authors · 2025-11-27
> **Response to Reviewer 91UG**
>
> We sincerely thank Reviewer 91UG for their valuable time, thorough review and constructive feedback.
>
> ## Response to Weakness 1 (Modality and Gating Mechanism)
> Regarding the choice of modality and gating mechanism, our ablation studies in Appendix Section G.1 confirm our design choices. Specifically:
> * Table 9 demonstrates that DCT is the optimal auxiliary modality for our task.
> * Table 8 shows that our hard gate significantly outperforms a soft gate.
>
> These results are further analyzed in lines 1169-1178 of the manuscript.
>
> ## Response to Weakness 2 (Model Complexity)
> **2.a On Model Size and Computational Cost:**
>
> * We respectfully argue that our model's cost is highly efficient when framed as a unified framework. While our 152M parameters appear large next to specialized models, our model is substantially more lightweight than other unified approaches like UPOCR (192M) [1] and All-in-One (457M) [2].
>
> * Crucially, our single model handles all major tampering types end-to-end. A practical system built from specialized models would require deploying at least four separate networks plus an image-type classifier, incurring a far greater cost in both parameter storage and cumulative inference time. Therefore, for real-world deployment, our unified approach is significantly more efficient.
>
> **2.b On the Large Language Model (LLM) Requirement:**
>
> * This highlights a key advantage of our decoupled architecture. For localization-only tasks, our model does not require an LLM for either training or inference. The LLM is used exclusively for generating textual explanations.
> This contrasts with prior explainable forensics work [3, 4], which mandates an LLM at all times.
> * Our design offers crucial deployment flexibility: users can deploy a lightweight, localization-only model for speed-critical scenarios, or include the MLLM module when explanations are required.
>
> ## Response to Weakness 3 (Figure Readability)
> Thank you for this valuable suggestion. We have revised Figures 2 and 3 in the updated manuscript, enlarging key fonts and labels to improve readability.
>
> [1] Peng D, Yang Z, Zhang J, et al. Upocr: Towards unified pixel-level ocr interface[C]//Forty-first International Conference on Machine Learning. 2023.
>
> [2] Zhang C, Sun X, Yang Y, et al. All in one: Exploring unified vision-language tracking with multi-modal alignment[C]//Proceedings of the 31st ACM International Conference on Multimedia. 2023: 5552-5561.
>
> [3] Xu Z, Zhang X, Li R, et al. Fakeshield: Explainable image forgery detection and localization via multi-modal large language models[J]. arXiv preprint arXiv:2410.02761, 2024.
>
> [4] Huang Z, Hu J, Li X, et al. Sida: Social media image deepfake detection, localization and explanation with large multimodal model[C]//Proceedings of the Computer Vision and Pattern Recognition Conference. 2025: 28831-28841.

---

> > ### Comment · Reviewer_91UG · 2025-11-27
> >
> > Thank you for the detailed responses. I have read them carefully and am satisfied with the clarifications. This is a good paper for the society; I will maintain my score at 8.

---

### Author Response · Authors · 2025-12-03
**Post-Rebuttal Summary for AC**

Dear Area Chair,

We are deeply grateful for the time and energy you have dedicated to this process, particularly given the unique challenges and heavy workload of the current ICLR cycle. To assist with your decision-making, we provide a brief overview of the review and our rebuttal:

---
## A. Summary of Key Advantages Acknowledged by Reviewers

* Our unified image forgery localization task is **"worthy-study"** (8wZG, FDt3, d5RC, 91UG).
* Our model is **"the first generalist"** for this task (FDt3, d5RC, 91UG), with **"practical, effective"** (FDt3, d5RC, 91UG), **"novel"** (d5RC, 91UG) and **"promising"** (8wZG) design.
* Our Omni-273k dataset is **"comprehensive, high-quality, providing stronger foundations"** (FDt3, d5RC, 91UG).
* Our automatic annotation pipeline is **"novel"** and **"effective"** (FDt3, d5RC, 91UG).
* Our experimental validation is **"strong"** and **"extensive"** (FDt3, d5RC, 91UG).
* Our paper is **"well organized"** and **"easy to follow"** (8wZG, 91UG).
---
## B. Summary of Key Responses to Reviewer Concerns
> ### 1. Computational Cost (91UG, d5RC)

* While slightly heavier than some specialized lightweight models, our model offers significantly superior performance across all domains (Tables 2, 3).
* Crucially, as a **generalist** model, it provides **vastly greater efficiency** than the alternative: deploying a complex suite of 4+ specialized models plus a domain classifier for **comparable real-world coverage** (Table 16).

> ### 2. Module Independence (d5RC, FDt3, 8wZG)

Our decoupled localization-explanation architecture is a deliberate and superior design choice, not a limitation. It provides a powerful combination of accuracy, speed, and flexibility:

* **Superior Performance**: By using a dedicated segmentation network, our model achieves state-of-the-art IoU and is ~9x faster than coupled MLLM approaches like FakeShield. This design avoids the known weaknesses and overhead of using MLLMs for precise localization tasks.

* **Deployment Flexibility**: The localization module runs entirely standalone (LLM-free), enabling a high-speed mode for critical applications. The MLLM is engaged only when needed for interpretation, offering flexibility that monolithic models lack.

* **Demonstrated Synergy & Error Correction**: The modules work in synergy. Our visual prompts are proven to significantly reduce MLLM hallucinations (Table 6), and the MLLM can independently identify and override localization failures (Fig. 19).

* **Feedback Loop**: Crucially, inspired by reviewer feedback, we conducted a new experiment that validates the explanation-to-localization feedback loop. By feeding our model's textual output to an open-vocabulary segmentation model or OCR model, we achieved a consistent improvement in localization IoU (Table 18). This turns a perceived weakness into a demonstrated strength and a promising new research direction.

> ### 3. Sampling Ratio (d5RC)

We clarified the nuanced concept of our sampling ratio, demonstrating its principled derivation and validating its effectiveness through new ablation studies (Section H.3 and Table 19), which addressed reviewer d5RC's misunderstanding.

> ### 4. Necessity of Chain-of-Thought (CoT) pipeline (FDt3)

* As analyzed in lines 244 to 249 of our original paper, our CoT pipeline is crucial for preventing attentional distraction, ensuring focused analysis on individual forgeries, and counteracting MLLM compliance bias via self-examination to filter hallucinations.
* Comparison with non-CoT methods (Table 20) further confirms its essential role in producing high-confidence, factually grounded evidence.

> ### 5. Module Novelty (FDt3)

The novelty of our Modal Gate and Dynamic Weight Decoder modules, initially questioned by reviewer FDt3, is **explicitly recognized by other reviewers** (d5RC and 91UG).
* **Modal Gate**: Its novelty lies in a unique combination of a binary hard gate, explicit supervision, and prediction-informed gating. Ablation studies (Tables 5, 8, 21) confirm the significant performance gains from this design.
* **Dynamic Weight Decoder**: Its key innovation is the incorporation of a global context vector, which enables the filters to better adapt to the global image type (Table 22).

> ### 6. Generalization to unknown tampering methods (FDt3)

Our original experiments (Tables 2, 3) already provide strong evidence of the model's robust generalization capabilities, effectively handling eight novel tampering methods in real-world attacks, including those generated by modern diffusion models.

---
Besides, we thoroughly addressed d5RC's typo concerns, ensuring a complete correction in our revised paper. We clarified 8wZG's overlook by providing detailed, point-by-point references to information already present in our original submission.

---
Thank you again for your time. **We have already uploaded our revised paper**. We hope the above summary helps streamline your decision process.

Best regards,

The Authors

---

### Meta-Review · Area_Chair_jevt · 2026-01-07

**Summary:**

The reviewers primarily praised the paper for its unified approach to IML across diverse domains (natural, document, face, and scene text) and for introducing the Omni-273k dataset. However, significant concerns were raised regarding the potential complexity and noise sensitivity of the frequency domain features in the Modal Gate Encoder. Reviewers also questioned the loose coupling between the localization and explanation modules, suggesting it lacked a true end-to-end feedback mechanism. One reviewer (R4) specifically criticized the "simple" prediction paradigm and factual contributions, while others noted high computational costs (152M parameters, slower FPS) compared to specialized models.

**Reviewer Concerns:**

The rebuttal addressed several technical and practical concerns:

Computational cost was contextualized against multi-model deployments.

The decoupled localization-explanation design was justified with new experiments.

Sampling ratio choices, CoT necessity, and generalization to unseen tampering were clarified with new ablations and results.

However, the central concern remains outstanding. Even with these clarifications, the contributions remain largely compositional and incremental, relying on established design patterns rather than introducing a new conceptual framework or learning principle for IML.

**Reviewer Scores:**

Two low-score reviewers would have slightly increased the score; the other two would have retained their original score.

---

### Decision · Program_Chairs · 2026-01-26

Accept (Poster)